# Position: Don't be Afraid of Over-Smoothing and Over-Squashing

## Abstract

Over-smoothing and over-squashing have been extensively studied in the literature on Graph Neural Networks (GNNs) over the past years. In this position paper, we challenge this prevailing focus in GNN research, arguing that these phenomena are less critical for practical applications than assumed. We suggest that performance decreases often stem from uninformative receptive fields rather than over-smoothing. We support this position with extensive experiments on several standard benchmark datasets, demonstrating that accuracy and over-smoothing are mostly uncorrelated and that optimal model depths remain small even with mitigation techniques, thus highlighting the negligible role of over-smoothing. Similarly, we challenge that over-squashing is always detrimental in practical applications. Instead, we posit that the distribution of relevant information over the graph frequently factorises and is often localised within a small $k$-hop neighbourhood, questioning the necessity of jointly observing entire receptive fields or engaging in an extensive search for long-range interactions. The results of our experiments show that architectural interventions designed to mitigate over-squashing fail to yield significant performance gains. This position paper advocates for a paradigm shift in theoretical research, urging a diligent analysis of learning tasks and datasets using statistics that measure the underlying distribution of label-relevant information to better understand their localisation and factorisation.

## 1. Introduction

A graph is a versatile data structure that is well-suited for describing complex systems, offering a mathematical foundation to analyse real-world systems using Graph Neural Networks (GNN). Gilmer et al. (2017) introduced the Message Passing Neural Network (MPNN) framework in 2017, which serves as the foundation for most modern GNNs, with powerful variants like Graph Convolutional Networks (GCNs) (Kipf & Welling, 2017) and Graph Attention Networks (GATs) (Veličković et al., 2018; Brody et al., 2022) achieving state-of-the-art results on many graph learning tasks. These advances have spurred a wide range of applications. For example, GNNs are now routinely applied in chemistry and drug discovery for molecular property prediction (Li et al., 2021), in large-scale recommendation engines in social networks (Ying et al., 2018; Zhao et al., 2025; Liu et al., 2024), and in spatial–temporal forecasting for traffic prediction like for Google Maps (Yu et al., 2017; Derrow-Pinion et al., 2021). Theoretical work on GNNs has explored several challenges that limit their capabilities for graph learning tasks. Most prominently, the problems of over-smoothing, over-squashing, expressivity, and robustness are discussed in many publications over the past eight years (Rusch et al., 2023a; Akansha, 2025; Zhang et al., 2025; Morris et al., 2019; Zhang et al., 2024; Günnemann, 2022). Over-smoothing occurs when node representations become indistinguishable after many layers. Over-squashing refers to the issue where node embeddings of constant size struggle to represent information of exponentially growing receptive fields. Robustness examines a GNN's sensitivity to perturbations in input features or graph structure, with research focusing on adversarial attacks and defences. Finally, expressivity investigates the theoretical power of GNNs to distinguish between different graph structures and learn complex functions, often explored by comparing them to variants of Weisfeiler-Lehman test. While addressing all four challenges remains an active focus of current GNN research, this paper critically examines over-smoothing and over-squashing.

**This position paper questions the practical relevance of over-smoothing and over-squashing in currently established learning tasks. We believe a diligent analysis of the learning tasks with statistics that measure the underlying distributions of relevant information is required. It is crucially important that theoretical problems presumed to be of practical impact are quantified in practice to guide future directions of theoretical research.**

We advocate for the use of such statistics to ensure the

[1]Anonymous Institution, Anonymous City, Anonymous Region, Anonymous Country. Correspondence to: Anonymous Author <anon.email@domain.com>.

Preliminary work. Under review by the International Conference on Machine Learning (ICML). Do not distribute.

practical relevance of the emerging research field of Graph Representation Learning. In a recent position paper, Bechler-Speicher et al. (2025) posit that this field may lose relevance due to poor benchmarks. They criticise (1) the relevance of learning tasks on current benchmark datasets, (2) the suitability of certain benchmark datasets for the application of GNNs, and (3) the overall benchmarking culture. Consequently, they propose a paradigm shift towards driving impactful advances in graph learning research and call for new transformative real-world applications. We anticipate that this will trigger an extensive search for such applications, potentially followed by the introduction of new datasets and learning tasks, such as the recently proposed GraphBench (Stoll et al., 2025). We posit that not only the relevance of the datasets, but also that of the investigated problems, is important for preserving the relevance of research in Graph Representation Learning. A phenomenon becomes a problem when its impact can be measured.

## 2. Preliminaries.

Let $\mathcal{G} = (\mathcal{V}, \mathcal{E})$ be an undirected graph with node set $\mathcal{V}$ and edge set $\mathcal{E}$. The graph $\mathcal{G}$ can be represented via an adjacency matrix $\boldsymbol{A} \in \mathbb{R}^{\mathcal{V} \times \mathcal{V}}$ with $a_{vu} \neq 0$ if and only if $(v, u) \in \mathcal{E}$. Each node $v \in \mathcal{V}$ has a neighbourhood $\mathcal{N}(v) = \{u : (v, u) \in \mathcal{E}\}$ of degree $d(v) = |\mathcal{N}(v)|$, a $k$-hop neighbourhood $\mathcal{N}^{(k)}(v) = \{u \in \mathcal{V} \mid \delta(v, u) \leq k\}$, where $\delta(v, u)$ denotes the shortest path distance between $v$ and $u$ in the graph $\mathcal{G}$, and features $\boldsymbol{h}_v^{(0)} \in \mathbb{R}^{d_0}$, which are the rows of a matrix $\boldsymbol{H}^{(0)} \in \mathbb{R}^{\mathcal{V} \times d_0}$. Graph Neural Networks (GNNs) are neural networks designed to process graph-structured data. Most GNNs fall into the class of MPNNs, in which layers of message passing functions $M(\cdot)$ and update functions $U(\cdot)$ are iteratively composed. In layer $\ell$, a new hidden node representation is computed as

$$\boldsymbol{h}_v^{(\ell+1)} = U\left(\boldsymbol{h}_v^{(\ell)}, M\left(\boldsymbol{h}_v^{(\ell)}, \{\boldsymbol{h}_u^{(\ell)} : u \in \mathcal{N}(v)\}\right)\right).$$

To exemplify this abstract function class, we consider the model equation

$$\boldsymbol{H}^{(\ell+1)} = \sigma\left(\tilde{\boldsymbol{A}}\boldsymbol{H}^{(\ell)}\boldsymbol{W}^{(\ell)}\right)$$

of the GCN (Kipf & Welling, 2017), where the update step is defined by the non-linearity $\sigma$ and a learnable weight matrix $\boldsymbol{W}^{(\ell)}$, and the message passing step is implemented via the message passing operator $\tilde{\boldsymbol{A}} = (\boldsymbol{D} + \boldsymbol{I})^{-1/2}(\boldsymbol{A} + \boldsymbol{I})(\boldsymbol{D} + \boldsymbol{I})^{-1/2}$. Note that we omit trainable bias vectors in this work for brevity. For a MPNN, the receptive field of a node $v$ refers to the set of nodes whose initial features $\boldsymbol{h}^{(0)}$ influence the final embedding $\boldsymbol{h}_v^{(L)}$, i.e., all nodes with $\delta(v, u) \leq L$.

## 3. Over-Smoothing

Over-smoothing refers to the phenomenon wherein node representations converge to a single representation with an increasing number of message passing steps and hence become uninformative. In many publications, this is assumed to be the main reason for a relatively small optimal number of message passing layers compared to other neural architectures, like state-of the art Convolutional Neural Networks (CNNs) with hundreds of layers (He et al., 2015; Tan & Le, 2021; Howard et al., 2019). We believe that the continued focus of theoretical research on over-smoothing within the graph learning community constrains the practical relevance of recent contributions. In this section, we substantiate our position by first reviewing the extensive body of literature on over-smoothing, which in our view indicates that the phenomenon is both well-understood and effectively mitigated. Based on the literature, we call into question the importance of working with arbitrarily many message passing layers in current GNN research. We support our position with experiments demonstrating that even GNNs designed to avoid over-smoothing do not yield significant performance gains on standard benchmarks.

### 3.1. Related Work

Kipf & Welling (2017) first discovered a performance peak of GNNs, here the Graph Convolution Network (GCN), at around 7–8 message passing layers. Li et al. (2018) further investigated this phenomenon based on the simplification of graph convolutions as a special form of Laplacian smoothing and showed that infinitely repeated applications of the message passing operator lead node embeddings to converge to a constant vector. Further, they demonstrated empirically for a small graph dataset (Zachary, 1977) that repeated graph convolutions first improve the class separation in the latent space before collapsing the representations.

Chen et al. (2019) approached the phenomenon empirically and were among the first trying to measure over-smoothing. They propose to use the cosine-based Mean Average Distance (MAD) and the MADGap as a quantitative metric for the smoothness and over-smoothness of the graph, respectively. Their experiments suggest a connection between over-smoothness, performance, and the information-to-noise ratio, quantified by the proportion of same-class nodes in an $L$-hop neighbourhood, where $L$ corresponds to the number of message passing layers.

Oono & Suzuki (2019) investigated the asymptotic behaviour of the expressive power of GCNs as the number of layers tends to infinity. They show that the node representations converge to a single signal determined by the node's connected component and degree, as well as the exponential nature of this asymptotic behaviour. They show that

$$d_{\mathcal{M}}(\boldsymbol{h}_v^{(\ell)}) \leq s\,\lambda\,d_{\mathcal{M}}(\boldsymbol{h}_v^{(\ell-1)}),$$

where $d_{\mathcal{M}}(\boldsymbol{h}_v^{(\ell)})$ denotes the distance of a node representation $\boldsymbol{h}_v$ in layer $\ell$ to the common representation in subspace $\mathcal{M}$ of the same connected component and node degree. This measures the smoothness of the graph. The over-smoothing factor $s\lambda$ comprises $\lambda$, the non-trivial eigenvalue of the propagation matrix with the largest absolute value, and $s$, the largest singular value of all weight matrices $\boldsymbol{W}^{(\ell)}$, or in other words, the maximum amplification factor. They argue that for most graphs, $s\lambda < 1$, implying inevitable over-smoothing.

Keriven (2022) provided a theoretical analysis of the effect of message passing, focusing not just on the asymptotic behaviour but also the optimal GNN depth. Using a stochastic latent-space random-graph model and analysing the mean-aggregation dynamics via the ergodic theorem, he showed that a limited number of message passing layers improves learning due to faster shrinking of non-principal than of principal components and compression of inter-communities variance before complete collapse occurs.

Building on the curvature-based framework for over-squashing introduced by Topping et al. (2022), Nguyen et al. (2023) connected positive edge curvature to over-smoothing. They showed that higher curvature $\kappa$ implies higher neighbourhood overlap between adjacent nodes

$$\frac{|\mathcal{N}(v) \cap \mathcal{N}(u)|}{\max(d(v), d(u))} \geq \kappa(v, u),$$

and hence contributes to the smoothing of node features.

Most recently, in a preprint, Keriven (2025) examined over-smoothing from an optimisation point of view. He introduces the idea of *backward over-smoothing* and argues that "as soon as the last layer of the GNN is 'trained', then gradients vanish at every layer and the GNN cannot train anymore", leading to spurious stationary points.

**Measuring Over-Smoothing.** Several measures for over-smoothing have been proposed in recent years, which differ primarily in two design choices: (1) the metrics used to quantify the distance between node representations, and (2) the pairs of nodes between which this distance is measured. The most prominent measure is the *Dirichlet energy*, which is based on squared differences between representations of neighbouring nodes (Rusch et al., 2023a; Cai & Wang, 2020). Wu et al. (2023) propose a *node similarity* measure over all node pairs, also based on squared differences. The *Mean Average Distance* (Chen et al., 2019) applies the cosine distance between each node and its neighbours within a given $k$-hop neighbourhood. We provide further details on

these measures in Appendix A. To measure over-smoothing in a GNN, the appropriate distance metric must be selected depending on the message passing scheme. The degree-normalised version of the Dirichlet energy (Cai & Wang, 2020) is expected to converge to zero for GCNs, as they collapse node representations to a constant vector scaled by the degree (Oono & Suzuki, 2019; Cai & Wang, 2020). The Dirichlet energy as used by Rusch et al. (2023a) and the node similarity (Wu et al., 2023) converge to zero if and only if all node representations become identical. This over-smoothing behaviour is expected for models with row-stochastic message passing operators like GATs. MAD can be applied to GCNs and GATs, due to the scale-invariance of the cosine distance.

**Methodologies Countering Over-Smoothing.** Several architectural interventions have been proposed to counter over-smoothing. Zhao & Akoglu (2020) introduced PairNorm to reduce the effect of over-smoothing by stabilising the sum of the distances between node pairs with and without edges. Chen et al. (2020) showed that simple skip connections can enable stable performance of deep GCNs. Specifically, they introduced GCNII, which adds the initial embedding to each hidden embedding by a proportion $\alpha$ and an identity mapping to each weight matrix $\boldsymbol{W}^{(\ell)}$ by a proportion $\beta$, where $\alpha$ and $\beta$ are hyperparameters. Scholkemper et al. (2025) proved theoretically that a GCN incorporating both batch-normalisation and residual connections does not suffer from over-smoothing. These theoretical findings are supported by empirical results on GNN with the most common message passing methods. Rusch et al. (2023b) further developed the idea of residual layers to gradient gating (G$^2$) by replacing the fixed proportion $\beta$ with learnable element-wise rates $\boldsymbol{\tau}^{(\ell)}$. In the G$^2$ model, $\boldsymbol{\tau}^{(\ell)}$ is given by a learnable function of the hidden features and the graph structure. Rusch et al. (2023a) surveyed these and other mitigation techniques and categorised them into (1) normalisation and regularisation, (2) change of the GNN dynamics, and (3) architectural enhancement through residual connections.

### 3.2. Position

We question the negative influence of over-smoothing in real-world datasets. The extensive work on over-smoothing has lead to a good theoretical understanding of this phenomenon. However, a lot of these works focus on proving an asymptotic behaviour. This work tries to put these theoretical results in the context of real-world datasets. We suspect the performance decrease on real-world data often arises not from over-smoothing, but as a consequence of *uninformative receptive fields*. For most learning tasks, a small receptive field is sufficient to obtain the relevant information. If the problem radius is smaller or similar to the number of message passing steps to achieve optimal smoothing, as

*Table 1.* Comparison of a GCN and three methods to mitigate over-smoothing for a range of model depths. The accuracy is given in % with the standard deviation in brackets. The best performing model depth is highlighted in bold. For two settings, our hardware ran out of memory (OOM).

| DATASET | MODEL | LAYERS | | | | | |
|---|---|---|---|---|---|---|---|
| | | 2 | 4 | 8 | 16 | 32 | 64 |
| CORA | GCN | **77.1 (3.3)** | 74.3 (0.4) | 36.0 (12.4) | 31.9 (0.0) | 31.9 (0.0) | 31.9 (0.0) |
| | GCN + PAIRNORM | **69.6 (5.5)** | 63.9 (5.5) | 60.4 (4.7) | 43.0 (9.1) | 36.4 (5.5) | 38.5 (9.1) |
| | GCNII | **76.3 (2.3)** | 75.3 (0.3) | 75.7 (0.5) | 75.6 (0.2) | 67.3 (17.7) | 76.3 (0.4) |
| | $G^2$ | 74.0 (2.4) | 75.2 (0.4) | 74.2 (0.4) | 75.0 (0.4) | **75.6 (1.1)** | 72.8 (4.0) |
| CITESEER | GCN | **75.5 (0.8)** | 74.5 (0.7) | 73.5 (0.4) | 58.1 (12.3) | 53.6 (19.2) | 18.1 (0.0) |
| | GCN + PAIRNORM | 68.2 (1.1) | **68.3 (1.6)** | 66.4 (4.2) | 63.6 (5.2) | 51.6 (5.8) | 21.9 (5.3) |
| | GCNII | 76.0 (0.7) | 76.4 (0.6) | 76.6 (0.6) | 77.4 (0.4) | 77.4 (0.4) | **77.6 (0.2)** |
| | $G^2$ | 75.9 (0.5) | **76.5 (0.4)** | 76.1 (0.4) | 74.8 (3.5) | 75.7 (0.5) | 76.1 (0.5) |
| PUBMED | GCN | **87.7 (0.4)** | 86.1 (0.2) | 83.0 (4.2) | 79.3 (5.6) | 63.2 (11.8) | 40.5 (0.4) |
| | GCN + PAIRNORM | **87.9 (0.4)** | 86.4 (0.3) | 85.8 (0.5) | 84.7 (0.3) | 84.7 (0.6) | 80.2 (3.0) |
| | GCNII | 88.3 (0.3) | 88.2 (0.4) | 88.4 (0.2) | 88.5 (0.2) | 88.2 (0.2) | **88.6 (0.2)** |
| | $G^2$ | **87.6 (1.1)** | 87.0 (1.5) | 86.8 (2.9) | 80.3 (20.9) | 80.2 (20.8) | 63.2 (28.6) |
| ROMAN-EMPIRE | GCN | **41.5 (0.7)** | 27.3 (1.2) | 18.7 (0.6) | 16.5 (0.3) | 15.0 (0.9) | 14.6 (0.6) |
| | GCN + PAIRNORM | **37.3 (0.4)** | 20.4 (0.7) | 19.6 (1.1) | 20.1 (1.2) | 15.5 (0.8) | 17.4 (3.3) |
| | GCNII | 56.9 (1.0) | **65.1 (0.6)** | 61.0 (0.9) | 60.3 (0.2) | 59.0 (0.2) | 58.0 (0.5) |
| | $G^2$ | 62.5 (3.3) | **62.9 (2.9)** | 60.6 (2.3) | 61.2 (3.3) | 61.0 (3.0) | OOM |
| OGBN ARXIV | GCN | **55.7 (0.2)** | 54.2 (0.5) | 32.1 (18.5) | 12.2 (9.0) | 7.2 (0.3) | 7.0 (0.8) |
| | GCN + PAIRNORM | 60.2 (0.3) | 60.3 (0.3) | **61.0 (0.2)** | 59.1 (0.3) | 53.2 (2.8) | 11.8 (4.7) |
| | GCNII | 58.2 (0.5) | **58.5 (0.6)** | 56.4 (2.5) | 56.9 (0.3) | 56.6 (0.4) | 56.2 (0.2) |
| PHOTO | GCN | 86.0 (4.5) | 73.7 (8.8) | 48.6 (10.0) | 29.9 (12.1) | 22.8 (4.4) | 25.9 (2.0) |
| | GCN + PAIRNORM | **87.0 (1.1)** | 78.0 (2.2) | 81.7 (2.2) | 81.4 (2.0) | 78.7 (5.3) | 65.0 (6.2) |
| | GCNII | **86.9 (2.0)** | 86.5 (1.7) | 85.5 (3.1) | 84.5 (4.5) | 85.0 (3.8) | 85.1 (5.1) |
| | $G^2$ | 78.3 (6.2) | 77.4 (10.7) | 82.9 (3.3) | **83.6 (3.2)** | 82.3 (3.3) | 77.2 (4.0) |
| COMPUTERS | GCN | **77.7 (3.7)** | 73.1 (4.5) | 54.4 (16.3) | 21.1 (14.7) | 24.4 (16.8) | 10.7 (12.0) |
| | GCN + PAIRNORM | **80.7 (2.4)** | 75.3 (3.2) | 76.0 (2.3) | 65.3 (17.8) | 45.5 (20.0) | 59.2 (11.1) |
| | GCNII | **81.5 (1.9)** | 81.3 (1.5) | 81.2 (1.7) | 75.1 (14.5) | 80.2 (3.9) | 81.2 (2.1) |
| | $G^2$ | 66.1 (9.9) | **74.6 (4.9)** | 73.3 (5.9) | 70.6 (9.8) | 74.5 (3.7) | OOM |

described by Keriven (2022), the performance will not suffer from over-smoothing. We support our position about the negligible effect of over-smoothing with experimental results in Section 3.4, which show that large model depths fail to gain significant performance improvements even if over-smoothing is prevented.

We furthermore want to specify that the phenomenon of over-smoothing is known to originate from the repeated application of message passing functions. Consequently, one can still fit *arbitrarily deep GNNs* without encountering over-smoothing by applying a limited number of message passing functions while working with arbitrarily deep update functions. Since, especially for large graph structures, the implementation of message passing comes at a significant computational expense, we posit that the desire to perform arbitrarily many message passing operations is not aligned with the practical deployment of GNNs.

Finally, we want to highlight that scaling the performance of GNNs to a very large number of message passing operations may be misguided due to the *methodological alternatives* that have been developed over the past years. Specifically, for problems with a very large problem radius, i.e., problems that justify the consideration of a very large receptive field

per node, one may almost surely want to explore alternatives such as Graph Transformers (Kreuzer et al., 2021; Ying et al., 2021; Rampášek et al., 2022) or rewiring techniques (Deac et al., 2022; Karhadkar et al., 2023; Arnaiz-Rodríguez et al., 2022) to consider these large receptive fields more efficiently than to perform as many or more message passing operations than the problem radius.

**Call to Action.** As a consequence of the questionable practical relevance of over-smoothing, the graph learning community should focus its efforts on phenomena of proven relevance implied by the conditions in given learning tasks and datasets. We suggest it might be helpful to develop statistics that quantify the label relevance of different-sized receptive fields. This quantification of the problem radius, in combination with a consideration of the optimal smoothing depth, should guide our efforts in improving model architectures.

### 3.3. Alternative Views

In parallel and independent work Arnaiz-Rodriguez & Errica (2026) challenge common beliefs by constructing theoretically sound counterexamples. In one of their po-

sitions, they question the performance-degrading effect of over-smoothing, which aligns with our arguments and experimental results. While we suspect uninformative receptive fields as a reason for limited performance in deep GNNs and call for measuring the relevance of label distributions, they emphasise the importance of node separability, which can be influenced by many effects and suggest further research efforts in this direction. Furthermore, Park et al. (2025) posit that "over-smoothing has been confused with the vanishing gradient" and call for research on the question of whether shallow optimal depth is inherent to GNNs, whereas we advocate for a more problem-oriented approach to allocate research efforts guided by statistics that measure if theoretical phenomena like over-smoothing result in practical problems.

### 3.4. Empirical Findings

We support our position on over-smoothing with experimental results. They show that mitigation techniques in most cases prevent models from over-smoothing, but still yield optimal performance with a small model depth.

We conducted those experiments on common node-classification datasets using a standard GCN and three models designed to mitigate over-smoothing, namely a GCN with PairNorm (Zhao & Akoglu, 2020), a GCNII (Chen et al., 2020), and a $G^2$-GCN (Rusch et al., 2023b). For each model–dataset combination, we performed hyperparameter optimisation and training, and then evaluated both the classification performance (Table 1) and relevant over-smoothing measures (Figure 1) for an exponentially increasing number of message passing layers, ranging from 2 to 64. Appendix A and B provide more details about the over-smoothing measures and the experimental set-up.

The results show clearly that over-smoothing is of negligible relevance in the examined datasets. For the GCN with an increasing number of message passing layers, we can observe a performance decrease in Table 1 as well as a decreasing MAD in Figure 1. A concurrency of these effects, like on Cora in the 8th layer or for PubMed in the 32nd and 64th layer, could indicate over-smoothing. However, on other datasets, like Roman-Empire and Photo, the drop in performance precedes that in MAD, which could be explained by an uninformative receptive field. In contrast, the GCNII and the $G^2$-GCN, which are theoretically capable of compensating for over-smoothing and uninformative receptive fields, exhibit none of these behaviours, maintaining relatively stable accuracy and MAD across layers. PairNorm at least reduces the effect, resulting in a more moderate decline. However, in most cases, the optimal performance is achieved with a shallow model. In those cases where the best accuracy is obtained with a large number of layers, the improvement is marginal and does not justify the increasing

computational cost. These findings are consistent across additional experiments using GATv2 and GraphSAGE, as detailed in Appendix C. We can conclude that although the mitigation techniques successfully prevent over-smoothing, they do not improve performance, which supports our position.

**A Remark on the Over-Smoothing Measures.** In Figure 1, we observe that the Dirichlet energy for the GCN does not exhibit the expected exponential decrease reported in other works (Rusch et al., 2023a; Park et al., 2025). These different experimental results may arise from conducting hyperparameter tuning and model training for the experimental study in this work. These findings are consistent with empirical works that show that a simple scaling of the weight matrices (Arnaiz-Rodriguez & Errica, 2026; Roth & Liebig, 2024) prevents a collapse of the Dirichlet energy.

## 4. Over-Squashing

Over-squashing describes the phenomenon that information of an exponentially growing receptive field can not be represented in a fixed size hidden dimension. We believe that continued theoretical work on over-squashing should be guided by measurable problems that limit the performance of GNNs in practical applications. In this section, we base our position on the literature, reviewing different perspectives on the phenomenon. We address these perspectives and, as a consequence, question the relevance of over-squashing for practical applications. We support our position with experimental results showing no consistent improvement of the accuracy after mitigating over-squashing.

### 4.1. Related Work

Over-squashing refers to the extensively studied phenomenon in which too much information is forced to pass through overly small node embeddings. If a node $v$ is to be affected by features of node $u$ at distance $k$, a GNN requires at least $k$ message passing steps. However, an increasing number of message passings results in an exponentially growing number of messages being sent, leading to a loss of information. In other words, over-squashing can be defined as the inability to losslessly compress a receptive field that grows with the depth of the network in fixed-sized node representations. As a consequence of over-squashing, features from neighbours in different $k$-hops can not be considered jointly at the central node, preventing the modelling of interaction effects. In detail, the phenomenon of over-squashing has been defined in various ways, focusing on different aspects of the phenomenon.

The concept of over-squashing was introduced by Alon & Yahav (2020) with the broadest definition of over-squashing. Based on the exponential growth of the recep-

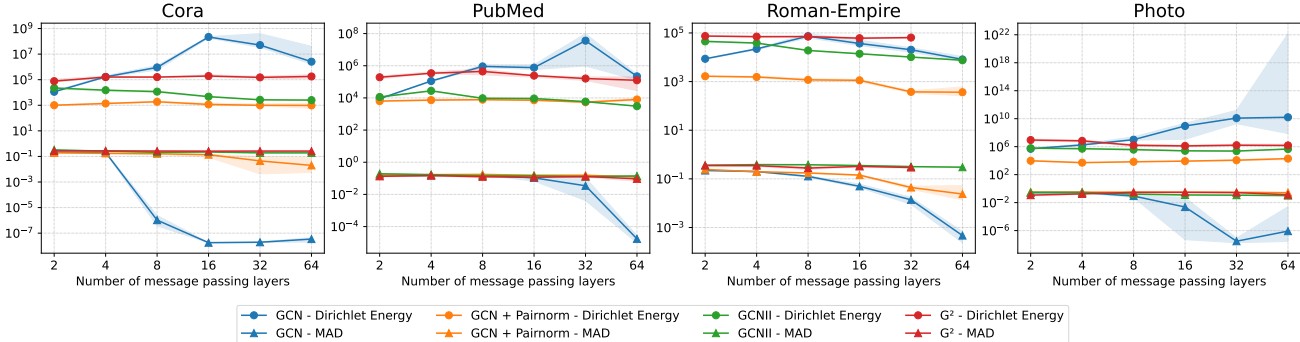

*Figure 1.* Median of over-smoothing measures of the whole dataset for all tested GCN variants and selected datasets (further results in Appendix A) at different depths after hyperparameter tuning and training. The error bands represent the 25- and 75-percentiles.

tive field of a node $v$ with the number of message passing $|\mathcal{N}^{(L)}(v)| = \mathcal{O}(e^L)$, they conclude an over-squashing of an exponentially growing amount of information while the embedding dimension remains constant. They emphasise the increasing severity of this phenomenon for problems that require *long-range information*, i.e., with a large problem radius $r$. With a synthetic dataset, Alon & Yahav (2020) showed the limitation of GNNs with the most common message passing methods for large problem radii and attributed this to over-squashing.

Topping et al. (2022) investigate over-squashing through *graph curvature*, focusing on bottleneck edges that induce over-squashing locally. They formulate the over-squashing phenomenon as an upper bound for the entries in the *Jacobian* and define it for the derivative of the hidden feature $h_v^{(r+1)}$ with respect to an input feature $h_u^{(0)}$ between a node $v$ and another node $u$ with $\delta(v, u) = r$ as

$$\left| \frac{\partial h_v^{(r+1)}}{\partial h_u^{(0)}} \right| \leq (\alpha \cdot \beta)^{r+1} \left( \tilde{\boldsymbol{A}}^{r+1} \right)_{v,u}, \qquad (1)$$

with the layer-wise Lipschitz constants $|\Delta U^{(\ell)}| \leq \alpha$ and $|\Delta M^{(\ell)}| \leq \beta$ for $0 \leq \ell \leq r$. The factor $(\tilde{\boldsymbol{A}}^{r+1})_{v,u}$ causes vanishing gradients, especially with bottleneck edges on the path between the two nodes. Topping et al. (2022) propose a modified Forman curvature, which uses the counts of triangles and 4-cycles based on an edge, to identify bottlenecks.

Di Giovanni et al. (2023) extended the fundamental understanding of over-squashing by investigating its impact on the width, depth and topology, viewing small magnitudes of Jacobian entries between features of a node $v$ and another node $u$ with a long-range interaction in general as a problem of over-squashing. They defined an alternative upper bound for the $L_1$-Norm of the Jacobian, which differs slightly from Equation (1) and is defined as

$$\left\| \frac{\partial \boldsymbol{h}_v^{(\ell)}}{\partial \boldsymbol{h}_u^{(0)}} \right\|_{L_1} \leq \underbrace{(c_\sigma \cdot w \cdot p)^\ell}_{\text{model}} \underbrace{\left( \tilde{\boldsymbol{A}}^\ell \right)_{v,u}}_{\text{topology}}, \qquad (2)$$

where $c_\sigma$ is the Lipschitz constant of the nonlinearity $\sigma$, $w$ is the maximal entry-value over all weight matrices, and $p$ is the hidden dimension or model width. They derive from Equation (2) that a larger hidden dimension prevents over-squashing if it compensates for the topology factor. However, they point out that an increased hidden dimension reduces the ability of generalisation and does not solve the actual problem, the graph topology. Further, they prove for tasks with long-range dependencies the occurrence of over-squashing for models with a depth comparable to the problem radius ($L \approx r$) and vanishing gradients for deeper models ($L \gg r$). Lastly, Di Giovanni et al. (2023) add another perspective on over-squashing and show that it occurs for a pair of nodes $v, u \in \mathcal{V}$ with a high commute time $\pi$, which measures the expected number of steps for a random walk to commute between $v$ and $u$.

**Methodologies Countering Over-Squashing.** Topping et al. (2022) introduce a curvature-based rewiring method incorporating their Balanced Forman metric to reduce over-squashing on bottlenecks. The *Stochastic Discrete Ricci Flow* (SDRF) adds edges to support low-curvature edges and removes high-curvature edges. A similar approach called *Batch Ollivier-Ricci Flow* (BORF) was proposed by Nguyen et al. (2023), using the Ollivier-Ricci curvature

$$\kappa(v, u) = \frac{W_1(\mu_v, \mu_u)}{\delta(v, u)},$$

where $W_1(\mu_v, \mu_u)$ is the $L_1$-Wasserstein distance and $\delta(v, u)$ is the shortest path distance. They use $\kappa$ for adding and removing edges in the graph to simultaneously mitigate

over-squashing and over-smoothing, respectively. Deac et al. (2022) create a modified graph structure using a fundamentally sparse family of expander graphs with a low diameter. In the Graph Expander Propagation (EGP), they replace $\mathcal{E}$ for every second propagation step with edges from the expander, reducing effective path lengths. Southern et al. (2025) build upon the observations of Di Giovanni et al. (2023) that over-squashing comes with long-range dependencies with long commute times and show theoretically and empirically that a virtual node can reduce $\pi$ and therefore reduce over-squashing.

### 4.2. Position

We challenge the presumed detrimental effect of over-squashing in real-world applications. More specifically, we question the practical relevance of joint observations of entire receptive fields, long-range interactions, and information exchange through bottlenecks.

The formulation of the over-squashing phenomenon through a limit of the Jacobian is based on the assumption that *long-range interactions* are generally informative for learning tasks and desirable to achieve through deep GNNs. We posit that in most practical applications on a reasonably constructed graph, the relevant information on interaction effects is stored within a small $k$-hop neighbourhood.

Next, we challenge the negative effect of low-curvature edges. *Bottleneck edges* limit the exchange of information between structural communities. An interrelation of negative curvature between communities and the label distribution has not been studied yet. However, we assume in most cases a correlation between these. Information exchange along bottlenecks is not relevant for such learning tasks.

At last, we consider the most general formulation of the over-squashing phenomenon, that is, the exponentially growing receptive field and the presumed impossibility of summarising its information in a constant hidden dimension. Implicit in this definition of over-squashing is the assumption that the information of the *entire receptive field* needs to be jointly observed to perform the given learning task. In other words, the impossibility definition states that the joint distribution over the receptive field of a node does not factorise into marginal distributions over the nodes, i.e., we assume the presence of high-order interaction effects within the receptive field of a node. We posit that this assumption is unrealistic and that for most real-world datasets, the joint distribution over the receptive field of a node does factorise and that we can hence process subsets of the receptive field independently and efficiently. In essence, it seems to us that a fixed-size node representation should be sufficient to successfully complete the majority of learning tasks for arbitrarily large receptive fields on real-world datasets.

**Call to Action.** We suggest that a diligent analysis is required to guide further theoretical research efforts and to better understand if real-world datasets cause the phenomenon of over-squashing and lead to problems in the application of GNNs. We call for the use of statistics to measure the localisation and factorisation of the feature and the label distributions.

To understand the importance of information exchange through structural bottlenecks and long-range dependencies, we call to measure the specific localisation of relevant node information with respect to a central node. In contrast to measuring the relevant problem radius, as discussed in Section 3.2, this notion of specific localisation must enable the identification of individual relevant nodes in order to draw conclusions about limitations arising from topological properties such as long-range dependencies and negative curvature. Initial attempts to quantify this have been made through the computation of the Jacobian (Bamberger et al., 2025; Liang et al., 2026). To meet the requirements of a specific localisation measure, this approach should be extended to detect outliers in the Jacobian rather than averaging over the $k$-hop neighbourhood, and to disentangle the overlapping effects induced by the model architecture from those inherent to the underlying data structure. Moreover, we suggest investigating the potential factorisation of the label distribution over $k$-hop receptive fields in order to assess the importance of jointly observing the entire receptive field.

The quantification of the different aspects of over-squashing has the potential to not only guide the development of future GNNs that are more closely aligned with the challenges posed by impactful, real-world learning problems, but also to offer an insightful categorisation of existing datasets and the ability of models to capture such effects. The application of these statistics shall allow us to better understand the practical implications of these phenomena and to set the right focus for further research directions.

### 4.3. Alternative Views

Arnaiz-Rodriguez & Errica (2026) criticise the current definition of over-squashing as inconsistent, which is compatible with our view. They decompose the over-squashing phenomenon into topological and computational bottlenecks, whereas we use a slightly different decomposition to structure our arguments. Our position on low-curvature edges aligns with their view on the relevance of topological bottlenecks in long-range tasks. While they call for synthetic datasets, like in Mathys et al. (2025), to investigate both types of bottlenecks separately, we call for research on measuring the different aspects of over-squashing in real-world datasets. Both share the same goal of better understanding these phenomena and their impact on GNNs.

*Table 2.* Comparison in graph-classification of a GIN and two methods to mitigate over-squashing for a range of model depths. The accuracy (or average precision for Peptides-func) is given in % with the standard deviation in brackets. The best performing model is highlighted in bold. The optimal depth per model and dataset is underlined.

| DATASET | MODEL | LAYERS | | | | | |
|---|---|---|---|---|---|---|---|
| | | **2** | **4** | **8** | **16** | **32** | **64** |
| MUTAG | GIN | 87.7 (3.1) | 86.3 (3.3) | 84.2 (2.2) | 86.7 (3.0) | 84.4 (3.9) | 88.2 (3.7) |
| | EGP (GIN) | **91.1 (1.4)** | 74.6 (11.2) | 61.1 (8.0) | 65.8 (2.3) | 63.9 (6.1) | 66.0 (3.5) |
| | GIN + BORF | 88.1 (2.7) | 86.5 (2.0) | 86.1 (1.7) | 87.0 (3.9) | 83.2 (4.9) | 83.5 (5.2) |
| | GIN + SDRF | 89.5 (2.6) | 86.3 (3.0) | 81.9 (3.0) | 82.3 (7.3) | 88.1 (3.9) | 86.5 (3.8) |
| ENZYMES | GIN | **60.6 (3.4)** | 59.0 (1.4) | 56.8 (3.2) | 47.7 (4.3) | 31.0 (3.6) | 17.7 (2.5) |
| | EGP (GIN) | 50.2 (2.7) | 45.3 (3.8) | 24.7 (3.5) | 18.8 (4.4) | 17.9 (3.3) | 16.6 (0.4) |
| | GIN + BORF | 56.9 (2.4) | 58.4 (2.1) | 57.6 (2.0) | 47.1 (3.0) | 30.5 (3.7) | 19.1 (2.9) |
| | GIN + SDRF | 50.3 (2.6) | 48.4 (2.3) | 44.6 (2.7) | 35.9 (3.0) | 27.0 (3.1) | 16.4 (1.8) |
| PROTEINS | GIN | 64.0 (1.1) | 64.3 (1.3) | 64.2 (1.3) | 63.9 (1.2) | 66.5 (2.2) | 66.1 (1.9) |
| | EGP (GIN) | 61.8 (0.7) | 65.6 (2.2) | 65.7 (3.4) | 68.1 (8.6) | 41.0 (15.5) | 46.8 (18.2) |
| | GIN + BORF | 65.3 (1.2) | 63.7 (1.0) | 72.6 (1.9) | 65.6 (2.1) | 66.9 (2.1) | 68.0 (1.4) |
| | GIN + SDRF | 70.3 (0.9) | 72.0 (1.3) | **72.9 (1.8)** | 72.8 (2.0) | 71.4 (2.0) | 68.1 (1.9) |
| PEPTIDES-FUNC | GIN | 59.3 (0.5) | **60.7 (1.0)** | 57.9 (1.1) | 60.7 (1.1) | 55.7 (1.2) | 42.2 (8.5) |
| | EGP (GIN) | 41.6 (0.3) | 42.9 (0.6) | 21.4 (3.2) | 17.5 (0.2) | 17.3 (0.3) | 17.2 (0.4) |
| | GIN + BORF | 59.0 (0.8) | 59.7 (0.6) | 57.9 (1.4) | 55.8 (1.4) | 49.7 (0.8) | 47.0 (1.4) |
| | GIN + SDRF | 56.7 (0.6) | 55.2 (0.8) | 53.1 (0.7) | 53.8 (1.0) | 49.8 (1.5) | 40.4 (4.1) |

*Table 3.* Regression results of one curvature value on the predictive performance for each dataset. Statistical significance is indicated using stars, with * denoting $p < 0.05$ and ** denoting $p < 0.005$.

| DATASET | BF CURVATURE | | OR CURVATURE | |
|---|---|---|---|---|
| | SLOPE | $p$-VALUE | SLOPE | $p$-VALUES |
| CORA | 0.011 | 0.441 | 0.026 | 0.455 |
| CITESEER | 0.004 | 0.248 | 0.018 | 0.256 |
| PUBMED | -0.002 | 0.553 | -0.003 | 0.552 |
| ROMAN-EMPIRE | 0.015 | 0.743 | 0.046 | 0.734 |
| PHOTO | -0.039 | 0.005** | -0.086 | 0.005** |
| COMPUTERS | -0.005 | 0.685 | -0.011 | 0.686 |
| MUTAG | 0.047 | 0.001** | -0.06 | 0.324 |
| ENZYMES | 0.344 | 0.009* | -0.143 | 0.09 |
| PROTEINS | -0.05 | 0.02* | 0.063 | 0.119 |
| PEPTIDES-FUNC | 0.028 | 0.562 | 0.255 | 0.027* |

### 4.4. Empirical Findings

In this section, we demonstrate that, in practical applications, there is no significant relationship between the reduction of measurable over-squashing and predictive performance.

We investigate the potential of over-squashing mitigation techniques, namely EGP (Deac et al., 2022), BORF (Nguyen et al., 2023), and SDRF (Topping et al., 2022), in combination with various message passing methods on common node-level and graph-level tasks. We performed hyperparameter optimisation, training and evaluation of the performance (Table 2) as described in Section 3.4. Furthermore, we evaluate the Balanced Forman (BF) and Ollivier-Ricci (OR) curvature discussed in Section 4.1 on the original graphs and the graphs modified by the mitigation techniques. This currently constitutes the only available method to quantify over-squashing. To assess the impact of curvature reduction on predictive performance, we fit a linear regression for each dataset, using accuracy as the dependent variable and the curvature as the independent variable (Table 3). Appendix D further details the curvature evaluation.

Considering the results in Table 2, we observe that neither the individual approaches nor graph rewiring in general consistently improves model accuracy. Further results for all message passing methods and datasets are provided in Appendix C. Moreover, as shown in Table 3, for none of the tested datasets are both estimated slope coefficients of curvature on predictive performance significantly greater than zero. Hence, we can conclude that local over-squashing through structural bottlenecks does not significantly limit predictive performance on common benchmark datasets.

## 5. Conclusion

In this position paper, we highlight the importance of an in-depth understanding of real-world learning problems to guide future research directions in theoretical work in graph representation learning. While much excellent work has been done to better understand and solve over-squashing and over-smoothing, the practical relevance of these phenomena should be questioned. As Bechler-Speicher et al. (2025) recently criticised the current benchmarking culture and called for new, transformative, real-world benchmarking datasets, we hope that many new learning tasks with individual challenges will emerge and gain importance. It seems crucial to us that, as part of this search for new applications and datasets, we analyse these diligently, especially for topics like over-smoothing and over-squashing. Statistics should be established to measure the localisation and factorisation of the feature, structure and label distributions.

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

# A. Over-Smoothing Measures

In the literature, several measures for over-smoothing have been proposed. In our experiments, we compute all of these measures to ensure that our conclusions are drawn from a comprehensive analysis.

Most prominently, the discrete Dirichlet energy is used to quantify converging node representations. For a feature matrix with a single feature or hidden dimension $\boldsymbol{x} \in \mathbb{R}^{|\mathcal{V}| \times 1}$, its vectorised form is defined as

$$E(\boldsymbol{x}) = \boldsymbol{x}^T \boldsymbol{L} \boldsymbol{x} = \frac{1}{2} \sum_{i=1}^{|\mathcal{V}|} \sum_{j=1}^{|\mathcal{V}|} a_{ij} \left( x_i - x_j \right)^2,$$

where $\boldsymbol{L}$ denotes the normalised Laplacian matrix of the graph. In multidimensional space, the Dirichlet energy is defined for a continuous function $u(\cdot)$ using the Euclidean norm $\| \cdot \|_2$ over all dimensions $c \in C$ by

$$E(u) = \frac{1}{2} \int_\Omega \|\nabla u\|_2^2 dx.$$

In our case, this corresponds to summing the Dirichlet energy over all feature dimensions, with $\|\boldsymbol{x}_i - \boldsymbol{x}_j\|_2^2 = (x_{ic} - x_{jc})^2 + \cdots + (x_{iC} - x_{jC})^2$. Since we compare the Dirichlet energy of hidden features across models with potentially different optimal feature dimensions, we normalise by the feature dimension to ensure comparability:

$$E(\boldsymbol{X}) = \frac{1}{C} \mathbf{1}^T \mathrm{diag}(\boldsymbol{X}^T \boldsymbol{L} \boldsymbol{X}) = \frac{1}{2C} \sum_{c=1}^{C} \sum_{i=1}^{|\mathcal{V}|} \sum_{j=1}^{|\mathcal{V}|} a_{ij} \left( x_{ic} - x_{jc} \right)^2. \tag{3}$$

Rusch et al. (2023a) define a slight variation of the Dirichlet energy for measuring over-smoothing on graphs, normalising by the number of nodes rather than by two. They also apply a square root to the Dirichlet energy. Note that both the normalisation and the square root merely rescale the values and do not affect the expected exponential decay towards zero. We therefore adopt the definition in Equation (3).

The Dirichlet energy is equal to zero if and only if all node representations are identical. However, this behaviour only arises after over-smoothing for example for GAT layers (Wu et al., 2023; Cai & Wang, 2020). For GCNs, over-smoothed representations are expected to be scaled by the node degree (Oono & Suzuki, 2019). Consequently, Cai & Wang (2020) propose a degree-normalised Dirichlet energy that converges to zero for over-smoothing in GCNs. Extending this to our multidimensional formulation yields

$$E(\boldsymbol{X}) = \frac{1}{C} \mathbf{1}^T \mathrm{diag}(\boldsymbol{X}^T \boldsymbol{L} \boldsymbol{X}) = \frac{1}{C} \sum_{c=1}^{C} \sum_{i=1}^{|\mathcal{V}|} \sum_{j=1}^{|\mathcal{V}|} a_{ij} \left( \frac{x_{ic}}{\sqrt{1 + d(i)}} - \frac{x_{jc}}{\sqrt{1 + d(j)}} \right)^2. \tag{4}$$

Wu et al. (2023) define a node similarity measure based on deviations from the global mean feature vector:

$$\mu(\boldsymbol{X}) = \left\| \boldsymbol{X} - \frac{\mathbf{1}\mathbf{1}^T \boldsymbol{X}}{|\mathcal{V}|} \right\|_F, \tag{5}$$

where $\| \cdot \|_F$ denotes the Frobenius norm. This metric also converges to zero in cases where GAT-based models exhibit over-smoothing.

Chen et al. (2019) introduce the Mean Average Distance (MAD), a smoothness metric based on the cosine similarity between pairs of nodes within a target mask. In our work, we compute only the neighbourhood smoothness, defined for a given $k$-hop neighbourhood as

$$\mu_{\mathrm{MAD}}^{(k)}(\boldsymbol{X}) = \frac{1}{|\mathcal{E}^{(k)}|} \sum_{i=1}^{|\mathcal{V}|} \sum_{j=1}^{|\mathcal{V}|} a_{ij}^{(k)} \left( 1 - \frac{\boldsymbol{x}_i \boldsymbol{x}_j}{\|\boldsymbol{x}_i\|_2 \|\boldsymbol{x}_j\|_2} \right),$$

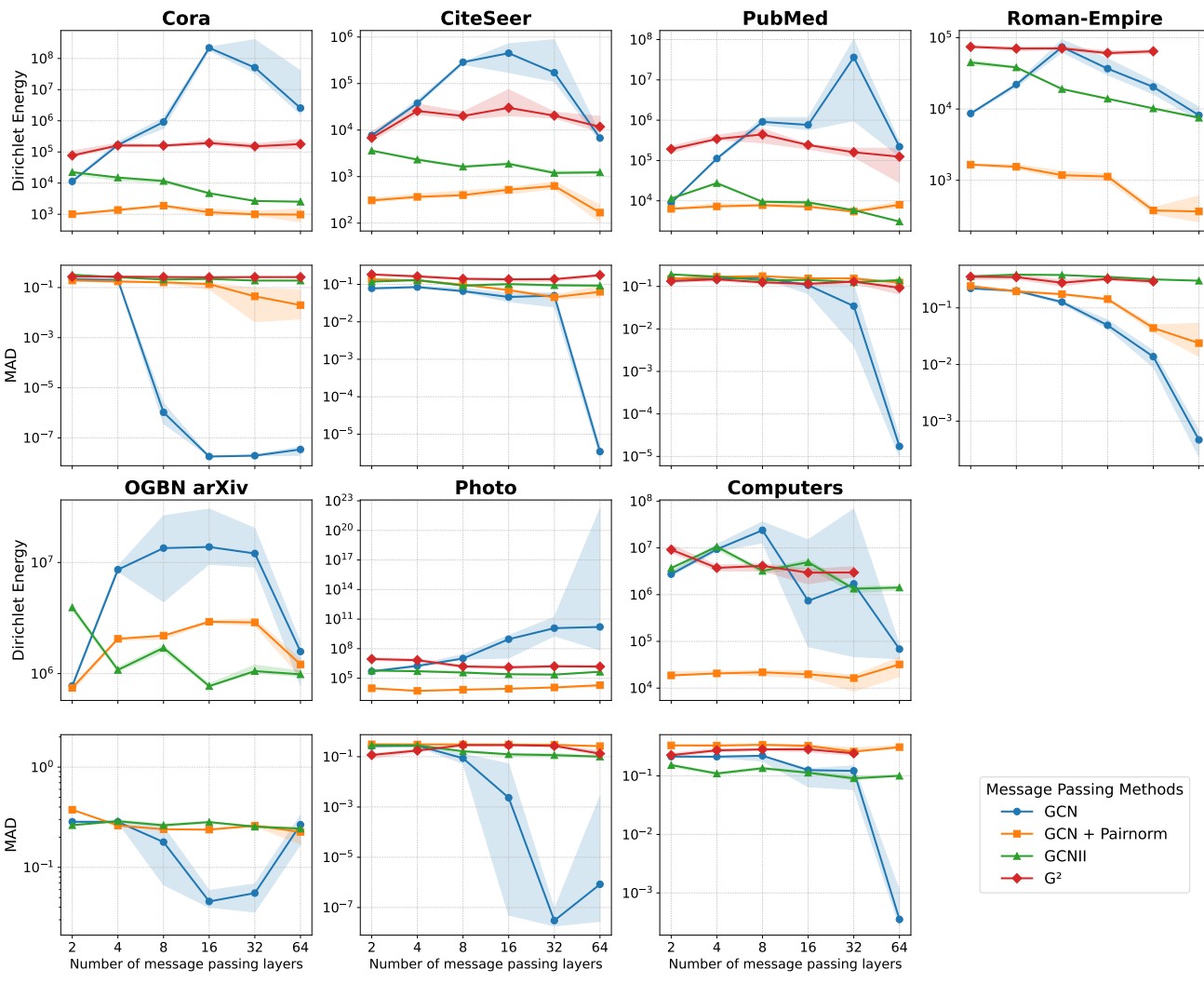

*Figure 2.* Dirichlet energy and MAD with a 3-hop neighbourhood of the whole dataset for all tested models with GCN message passing and datasets at different depths after hyperparameter tuning and training. The error bands represent the 25- and 75-percentiles.

with $\mathcal{E}^{(k)}$ and $a_{ij}^{(k)}$ are the edge set and entry of the adjacency matrix of the $k$-hop neighbourhood $\mathcal{N}^{(k)}(\cdot)$, respectively.

In this work, we evaluate all discussed over-smoothing measures on the hidden node embeddings after the final message passing and update step. We compute each measure for every random test initialisation, and for all nodes of the graph, not only those in the test set. In Figure 1 and in Figures 2 to 4, we report the measures theoretically appropriate for the respective model type: the degree-normalised Dirichlet energy from Equation (4) and MAD with $k = 3$ for models based on GCN message passing, and the Dirichlet energy from Equation (3), the node similarity measure from Equation (5), and MAD with $k = 3$ for models using GATv2 or GraphSAGE message passing.

## B. Experimental Details

The empirical results discussed in Sections 3.4 and 4.4 were obtained through an extensive set of experiments, which will be further detailed in this appendix.

We evaluated methods against over-smoothing on seven datasets from different domains and of varying sizes. We performed experiments on the citation datasets Cora, Citeseer, and PubMed (Yang et al., 2016), as these are the benchmarks on

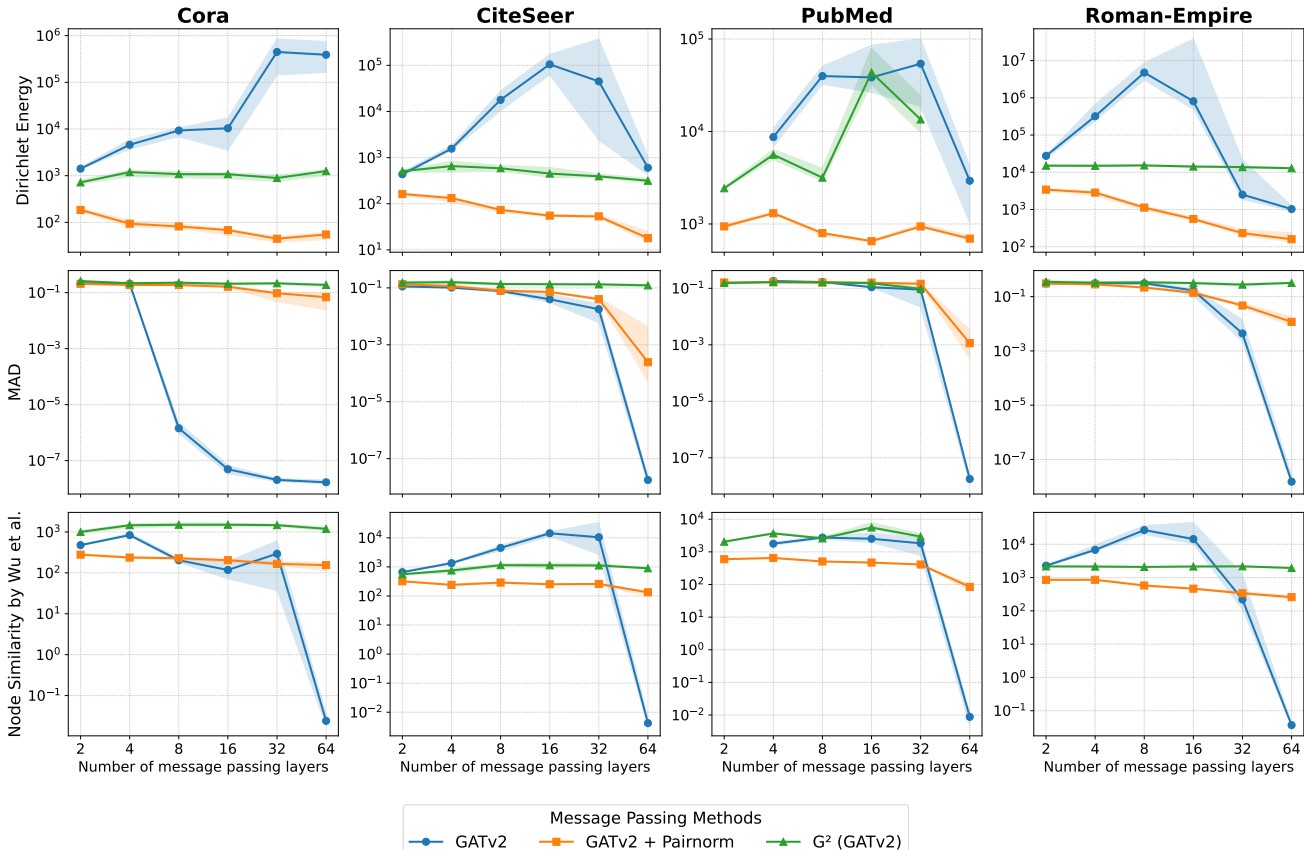

*Figure 3.* Dirichlet energy, MAD with a 3-hop neighbourhood and node similarity by Wu et al. (2023) of the whole dataset for all tested models with GATv2 message passing and datasets at different depths after hyperparameter tuning and training. The error bands represent the 25- and 75-percentiles.

which most over-smoothing mitigation methods have been developed. To further strengthen our experimental conclusions, we additionally included the citation network OGBN-arXiv (Hu et al., 2020), which is orders of magnitude larger, the heterophilic Roman-Empire text graph (Platonov et al., 2024), and the Photo and Computers co-purchase graphs from the Amazon dataset (Shchur et al., 2018). We did not include any datasets with graph-level tasks for the over-smoothing experiments, as this phenomenon has primarily been studied in the context of node-level tasks. For graph-level tasks, a performance degradation due to the convergence of node representations is also not expected, since hidden node embeddings are aggregated into a single graph representation after message passing.

For the experiments on over-squashing, we included four datasets with graph-level tasks in addition to the seven node-level datasets listed above. We used the datasets Mutag, Enzymes, and Proteins from the TU collection (Morris et al., 2020), as well as the Peptides-func dataset from the Long Range Graph Benchmark (Dwivedi et al., 2023). The former cover a range of typical domains for graph-level GNN applications, namely molecules, enzymes, and proteins. The latter dataset was chosen to validate our findings on a substantially larger benchmark. Further, the quality of the Peptides-func dataset as a benchmark was rated much higher (Coupette et al., 2025).

**A Remark on the RINGS Framework.** The good work of Coupette et al. (2025) is related to our call for statistics that quantify properties of benchmark datasets to preserve the practical relevance of research in Graph Representation Learning. However, in contrast to Coupette et al. (2025), who introduce a method to measure the quality of a dataset as a graph learning benchmark, we call for a systematic investigation of inherent problems within the learning tasks.

For each model, we optimised the hyperparameters on each dataset and for each number of message passing layers

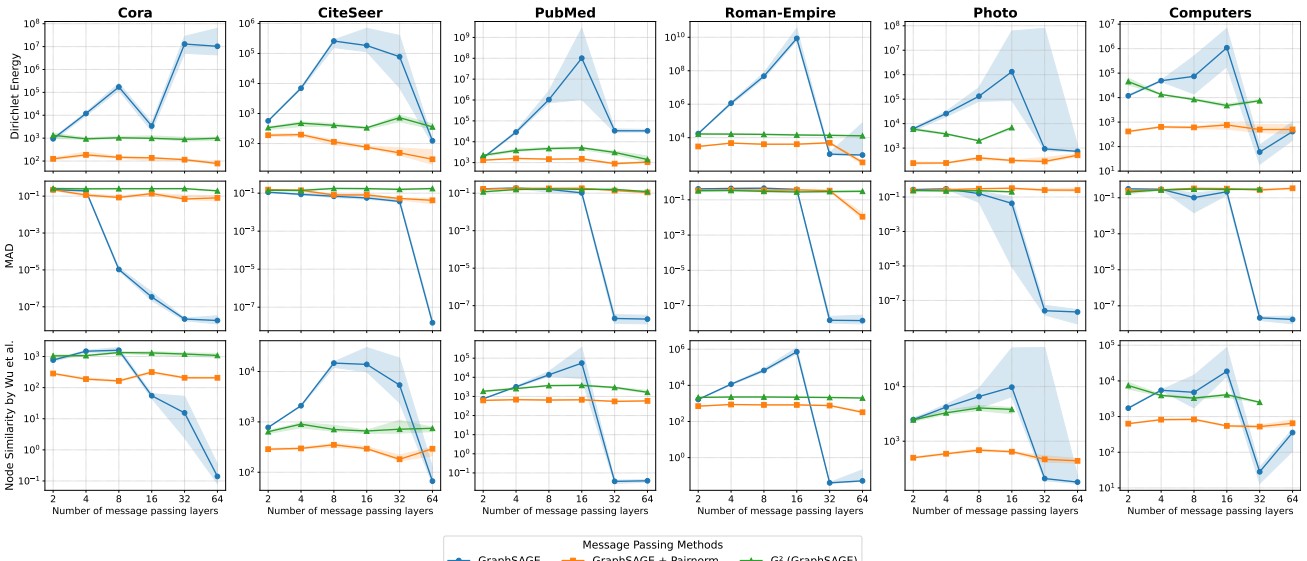

*Figure 4.* Dirichlet energy, MAD with a 3-hop neighbourhood and node similarity by Wu et al. (2023) of the whole dataset for all tested models with GraphSAGE message passing and datasets at different depths after hyperparameter tuning and training. The error bands represent the 25- and 75-percentiles.

$\ell \in \{2, 4, 8, 16, 32, 64\}$. The common hyperparameters, together with their search spaces or fixed values used for all models, are summarised in Table 4. In addition, Table 5 lists the hyperparameters specific to BORF and SDRF. For BORF, we defined the search spaces based on the optimal hyperparameters reported for Cora, Citeseer, PubMed, Mutag, Enzymes, and Proteins in the original paper (Yang et al., 2016). The search spaces for SDRF on Cora, Citeseer, and PubMed follow the optimal hyperparameters reported in Topping et al. (2022). For GCNII and $G^2$, the search spaces are not dataset-specific. We searched over $\alpha \in \{0.1, 0.3, 0.5\}$ and $\lambda \in \{0.5, 1.0, 1.5\}$ for GCNII, following Chen et al. (2020), and over the exponent $p \in \{1, 3, 5\}$ for $G^2$, following Rusch et al. (2023b).

The hyperparameter optimisation and testing procedures were implemented in accordance with the corresponding literature:

- **Cora, Citeseer, and PubMed.** For the three citation networks, we used the publicly available "planetoid" split (Yang et al., 2016). For hyperparameter optimisation, we combined the train and validation sets and performed a 10-fold cross-validation. The test set remained unchanged.

- **OGBn-arXiv.** For the arXiv citation network, we used the publicly available split (Hu et al., 2020). We combined the train and validation sets and performed a 8-fold cross-validation for hyperparameter optimisation. The test set remained unchanged.

- **Roman-Empire.** For the heterophilic Roman-Empire dataset, we used the publicly available split consisting of 10 random 50%/25%/25% train/validation/test splits (Platonov et al., 2024). We adjusted the testing procedure slightly and trained and tested the model with the optimal hyperparameter setting per fold with three random parameter initialisations as suggested by Errica et al. (2020).

- **Amazon.** For the Amazon Photo and Computers datasets, we applied the 20/30/rest split described in Shchur et al. (2018). Since the original splits are not publicly available, we implemented the sampling strategy, selecting 20 and 30 samples per class for the train and validation sets, respectively. For hyperparameter optimisation, we resampled eight stratified folds from the union of train and validation sets and conducted cross-validation. The test set from the original 20/30/rest split was kept fixed.

- **Mutag, Enzymes, and Proteins.** For the three TU datasets (Morris et al., 2020), originally published by Ivanov et al. (2019), we used the publicly available 80%/10%/10% train/validation/test splits. For both hyperparameter optimisation and testing, we followed the protocol of Errica et al. (2020).

*Table 4.* Search space or fixed values for standard hyperparameters defined for each dataset applied to every model; RoP($p$, $lr$, $f$, $lr_{min}$) denotes a starting learning rate $lr$ to be reduced by factor $f$ when the validation loss plateaus with patients $p$ down to a minimum of $lr_{min}$.

| DATASET | HIDDEN SIZE | DROPOUT | LEARNING RATE | EPOCHS (MAX) | PATIENCE | BATCH SIZE |
|---|---|---|---|---|---|---|
| CORA | {32, 64, 128} | {0.2, 0.4, 0.6, 0.8} | 0.001 | 500 | 30 | 1 |
| CITESEER | {32, 64, 128} | {0.2, 0.4, 0.6, 0.8} | 0.001 | 500 | 30 | 1 |
| PUBMED | {32, 64, 128} | {0.2, 0.4, 0.6, 0.8} | 0.001 | 500 | 30 | 1 |
| ROMAN-EMPIRE | {32, 64, 128} | {0.2, 0.4, 0.6, 0.8} | 0.001 | 500 | 30 | 1 |
| OGBN-ARXIV | {128, 256, 512} | {0.0, 0.2, 0.4} | RoP(25, 0.003, 0.5, 7,4E-4) | 2000 | 50 | 1 |
| PHOTO | {256, 512, 1024} | {0.0, 0.1, 0.2, 0.4} | RoP(10, 0.001, 0.5, 2,4E-4) | 3000 | 20 | 1 |
| COMPUTERS | {256, 512, 1024} | {0.0, 0.1, 0.2, 0.4} | RoP(10, 0.001, 0.5, 2,4E-4) | 3000 | 20 | 1 |
| MUTAG | {32, 64, 128} | {0.2, 0.4, 0.6, 0.8} | 0.01 | 2000 | 500 | 64 |
| ENZYMES | {32, 64, 128} | {0.2, 0.4, 0.6, 0.8} | RoP(5, 0.001, 0.5, 1E-6) | 500 | 50 | 64 |
| PROTEINS | {32, 64, 128} | {0.2, 0.4, 0.6, 0.8} | RoP(5, 0.001, 0.5, 1E-6) | 500 | 50 | 64 |
| PEPTIDES-FUNC | {128, 256, 512} | {0.0, 0.1, 0.2} | RoP(5, 0.003, 0.5, 1E-5) | 500 | 20 | 256 |

*Table 5.* Search space or fixed values for hyperparameters of preprocessing techniques for mitigating of over-squashing defined for each dataset applied to every model.

| DATASET | BORF | | | SDRF | |
|---|---|---|---|---|---|
| | ITERATIONS $N$ | EDGES ADDED $h$ | EDGES REMOVED $k$ | ITERATIONS $N$ | TEMPERATURE $\tau$ |
| CORA | {2, 3} | {10, 20} | {10, 20} | {100, 150} | {10, 100} |
| CITESEER | {2, 3} | {10, 20} | {10, 20} | {100, 150} | {10, 100} |
| PUBMED | {2, 3} | {10, 20} | {10, 20} | {100, 150} | {10, 100} |
| ROMAN-EMPIRE | {2, 3} | {10, 20} | {10, 20} | {100, 150} | {10, 100} |
| PHOTO | {2, 3} | {50, 100} | {50, 100} | {100, 150} | {10, 100} |
| COMPUTERS | - | - | - | {100, 150} | {10, 100} |
| MUTAG | {1, 2} | {3, 4} | {1, 2} | {10, 15} | {10, 100} |
| ENZYMES | {1, 2} | {3, 4} | {1, 2} | {10, 15} | {10, 100} |
| PROTEINS | {1, 2} | {3, 4} | {1, 2} | {10, 15} | {10, 100} |
| PEPTIDES-FUNC | 2 | 12 | 4 | 25 | 100 |

- **Peptides-func.** For the Peptides-func dataset from the Long Range Graph Benchmark (Dwivedi et al., 2023), we used the publicly available 70%/15%/15% train/validation/test splits. For hyperparameter optimisation, we combined the train and validation sets and performed a 8-fold cross-validation. The test set remained unchanged.

For the datasets for which we did not apply the Errica protocol, we evaluated each model on the test set using the hyperparameter configuration that achieved the best mean validation loss across all folds, and we report the mean over 10 random parameter initialisations.

## C. Extended Report of Experimental Results

In Tables 6 to 8 we report the results for all experiments, including additionally GATv2 and GraphSAGE as message passing methods.

## D. Curvature Results

We investigate the impact of reducing negative graph curvature on predictive performance using a linear regression.

We evaluate the two most common graph curvature measures, namely Ollivier–Ricci curvature (Nguyen et al., 2023) and Balanced Forman curvature (Topping et al., 2022), for all graph modifications used in our experiments. These include the original graph, the expander graph employed in the EGP model, as well as the graphs rewired using BORF (Nguyen et al., 2023) and SDRF (Topping et al., 2022), in all variations arising from the possible combinations of hyperparameters, which we report in Appendix B. As BORF and SDRF aim to modify only a small number of edges, the total graph curvature does not change substantially through rewiring. Therefore, after evaluating the curvature on all edges, we sorte them in descending order by curvature and compute the mean of the top $n$ edges, i.e., the most severe bottleneck edges, where $n$ denotes the maximum number of edges edited by either rewiring method.

For the correlation analysis, we associated each best-performing model on each dataset across all numbers of layers, that

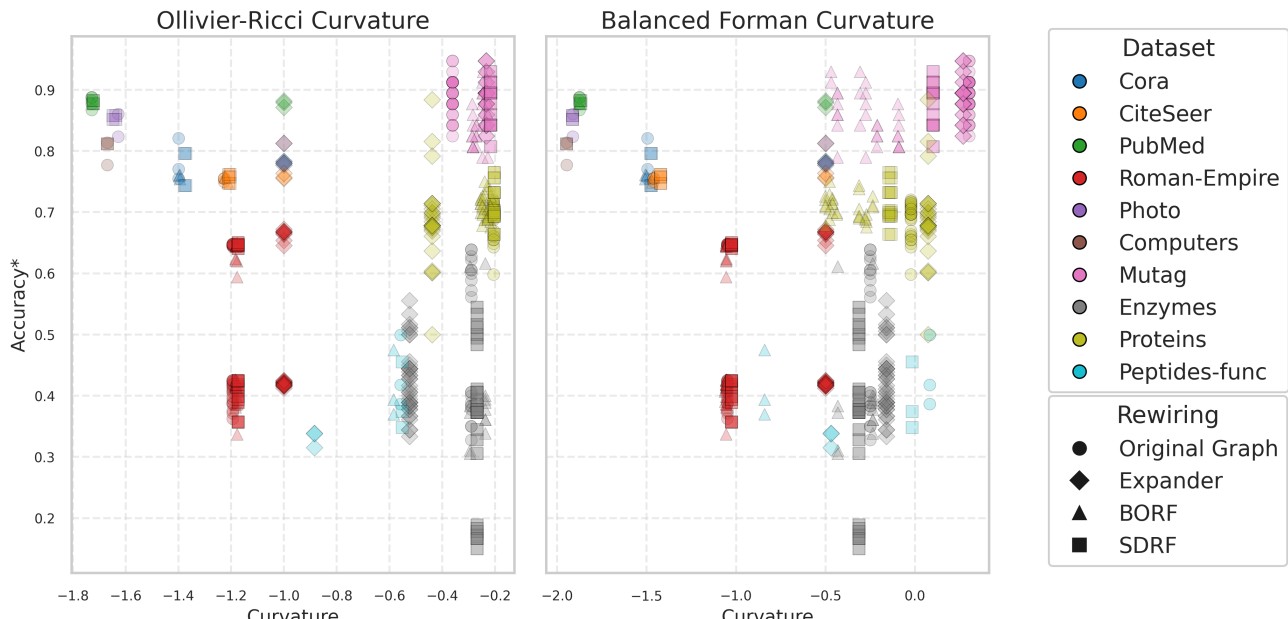

*Figure 5.* Accuracy (*or average precision) results for rewiring techniques to mitigate over-squashing with corresponding curvature values for the resulting message passing graph.

is, the scores underlined in Tables 2, 7 and 8, with the corresponding graph curvature for the method and hyperparameter configuration used to obtain the result. The resulting data points are visualised in Figure 5. Note that for the datasets Roman-Empire, Mutag, Enzymes, and Proteins, we obtain ten times as many data points per method, as these models are evaluated under the Errica protocol (Errica et al., 2020), which allows the hyperparameters to vary across random initialisations and therefore results in different curvature values for each test evaluation.

From Figure 5, we do not observe any apparent correlation between curvature and accuracy. To investigate this hypothesis quantitatively, we fit a linear regression for each dataset and rewiring method, with curvature as the independent variable and predictive performance as the dependent variable. This allows us to assess the impact of curvature on accuracy via the regression coefficient of the slope. To test whether this coefficient significantly differs from 0, we apply a standard $t$-test. The results of these regressions are reported in Table 3.

*Table 6.* Full table of experiments on node-level tasks with all tested models, comprising GCN and GATv2 and GraphSAGE in combination with tested mitigation techniques for over-smoothing for a range of model depths. The accuracy is given in % with the standard deviation in brackets. The best performing model is highlighted in bold.

| DATASET | MODEL | LAYERS | | | | | |
|---|---|---|---|---|---|---|---|
| | | **2** | **4** | **8** | **16** | **32** | **64** |
| CORA | GCN | **77.1 (3.3)** | 74.3 (0.4) | 36.0 (12.4) | 31.9 (0.0) | 31.9 (0.0) | 31.9 (0.0) |
| | GCN + PAIRNORM | **69.6 (5.5)** | 63.9 (5.5) | 60.4 (4.7) | 43.0 (9.1) | 36.4 (5.5) | 38.5 (9.1) |
| | GCNII | 76.3 (2.3) | 75.3 (0.3) | 75.7 (0.5) | 75.6 (0.2) | 67.3 (17.7) | 76.3 (0.4) |
| | $G^2$ | 74.0 (2.4) | 75.2 (0.4) | 74.2 (0.4) | 75.0 (0.4) | **75.6 (1.1)** | 72.8 (4.0) |
| | GATv2 | **75.5 (2.8)** | 73.9 (0.8) | 31.9 (0.0) | 31.9 (0.0) | 31.9 (0.0) | 31.9 (0.0) |
| | GATv2 + PAIRNORM | **69.7 (1.5)** | 61.5 (2.8) | 63.6 (6.4) | 61.8 (11.4) | 40.1 (8.0) | 38.8 (7.4) |
| | $G^2$ (GATv2) | 76.4 (1.7) | 77.7 (5.3) | 76.9 (3.3) | 75.0 (3.3) | 76.5 (4.1) | **78.5 (6.0)** |
| | GRAPHSAGE | **82.1 (5.1)** | 74.3 (0.5) | 31.9 (0.0) | 31.9 (0.0) | 31.9 (0.0) | 31.9 (0.0) |
| | GRAPHSAGE + PAIRNORM | **73.5 (4.9)** | 43.9 (6.4) | 37.9 (8.1) | 42.0 (3.5) | 36.2 (5.2) | 37.9 (5.1) |
| | $G^2$ (GRAPHSAGE) | 75.2 (0.5) | 75.2 (1.9) | 76.6 (2.1) | 77.9 (2.5) | 76.2 (1.7) | **78.6 (5.0)** |
| CITESEER | GCN | **75.5 (0.8)** | 74.5 (0.7) | 73.5 (0.4) | 58.1 (12.3) | 53.6 (19.2) | 18.1 (0.0) |
| | GCN + PAIRNORM | 68.2 (1.1) | **68.3 (1.6)** | 66.4 (4.2) | 63.6 (5.2) | 51.6 (5.8) | 21.9 (5.3) |
| | GCNII | 76.0 (0.7) | 76.4 (0.6) | 76.6 (0.6) | 77.4 (0.4) | 77.4 (0.4) | **77.6 (0.2)** |
| | $G^2$ | 75.9 (0.5) | **76.5 (0.4)** | 76.1 (0.4) | 74.8 (3.5) | 75.7 (0.5) | 76.1 (0.5) |
| | GATv2 | **75.0 (0.6)** | 74.1 (0.8) | 72.9 (0.6) | 58.7 (11.5) | 39.9 (8.4) | 21.1 (2.4) |
| | GATv2 + PAIRNORM | **68.4 (0.8)** | 66.5 (2.8) | 64.6 (4.9) | 63.6 (5.0) | 54.0 (5.5) | 21.1 (2.2) |
| | $G^2$ (GATv2) | 75.5 (3.6) | **76.3 (0.7)** | 76.0 (0.7) | 75.9 (0.5) | 76.3 (0.5) | 75.5 (0.5) |
| | GRAPHSAGE | **75.5 (0.4)** | 74.1 (0.4) | 72.8 (0.6) | 66.2 (4.7) | 49.9 (16.3) | 18.8 (0.0) |
| | GRAPHSAGE + PAIRNORM | **70.2 (1.0)** | 65.9 (4.8) | 56.3 (10.0) | 60.2 (7.3) | 34.5 (6.4) | 25.3 (7.8) |
| | $G^2$ (GRAPHSAGE) | **76.8 (0.3)** | 76.4 (0.5) | 76.2 (0.3) | 76.4 (0.5) | 76.6 (0.2) | 76.5 (0.5) |
| PUBMED | GCN | **87.7 (0.4)** | 86.1 (0.2) | 83.0 (4.2) | 79.3 (5.6) | 63.2 (11.8) | 40.5 (0.4) |
| | GCN + PAIRNORM | **87.9 (0.4)** | 86.4 (0.3) | 85.8 (0.5) | 84.7 (0.3) | 84.7 (0.6) | 80.2 (3.0) |
| | GCNII | 88.3 (0.3) | 88.2 (0.4) | 88.4 (0.2) | 88.5 (0.2) | 88.2 (0.2) | **88.6 (0.2)** |
| | $G^2$ | **87.6 (1.1)** | 87.0 (1.5) | 86.8 (2.9) | 80.3 (20.9) | 80.2 (20.8) | 63.2 (28.6) |
| | GATv2 | OOM | **86.7 (0.8)** | 84.7 (0.3) | 73.6 (2.0) | 63.1 (14.7) | 40.7 (0.0) |
| | GATv2 + PAIRNORM | **87.2 (0.5)** | 86.8 (0.5) | 84.7 (0.4) | 84.4 (0.4) | 82.3 (3.5) | 40.9 (0.5) |
| | $G^2$ (GATv2) | **88.2 (0.3)** | 86.6 (1.1) | 79.6 (20.6) | 71.6 (18.4) | 73.4 (20.7) | OOM |
| | GRAPHSAGE | 88.3 (0.4) | **88.6 (0.4)** | 81.4 (5.8) | 75.2 (3.8) | 40.7 (0.0) | 40.8 (0.2) |
| | GRAPHSAGE + PAIRNORM | 89.0 (0.3) | **89.6 (0.6)** | 89.2 (2.1) | 88.8 (0.3) | 80.7 (4.8) | 72.3 (3.1) |
| | $G^2$ (GRAPHSAGE) | **88.0 (0.3)** | 87.7 (0.3) | 86.9 (0.3) | 86.6 (1.5) | 86.4 (1.0) | 74.9 (23.4) |
| ROMAN-EMPIRE | GCN | **41.5 (0.7)** | 27.3 (1.2) | 18.7 (0.6) | 16.5 (0.3) | 15.0 (0.9) | 14.6 (0.6) |
| | GCN + PAIRNORM | **37.3 (0.4)** | 20.4 (0.7) | 19.6 (1.1) | 20.1 (1.2) | 15.5 (0.8) | 17.4 (3.3) |
| | GCNII | 56.9 (1.0) | **65.1 (0.6)** | 61.0 (0.9) | 60.3 (0.2) | 59.0 (0.2) | 58.0 (0.5) |
| | $G^2$ | 62.5 (3.3) | **62.9 (2.9)** | 60.6 (2.3) | 61.2 (3.3) | 61.0 (3.0) | OOM |
| | GATv2 | **39.2 (2.1)** | 38.5 (2.8) | 34.4 (3.7) | 19.3 (3.5) | 14.2 (0.3) | 13.9 (0.1) |
| | GATv2 + PAIRNORM | **52.5 (2.9)** | 33.6 (4.0) | 20.2 (1.6) | 18.0 (1.0) | 14.9 (0.4) | 14.1 (0.4) |
| | $G^2$ (GATv2) | 62.0 (3.6) | **63.2 (2.2)** | 61.8 (3.3) | 61.9 (2.4) | 61.1 (3.4) | 60.8 (2.4) |
| | GRAPHSAGE | **64.6 (0.3)** | 61.5 (2.6) | 46.5 (3.6) | 33.2 (3.1) | 13.9 (0.1) | 13.9 (0.1) |
| | GRAPHSAGE + PAIRNORM | 66.1 (0.6) | 65.6 (0.4) | **66.6 (0.5)** | 62.3 (1.5) | 35.8 (2.7) | 14.1 (0.4) |
| | $G^2$ (GRAPHSAGE) | **62.7 (3.4)** | 62.4 (2.3) | 60.2 (4.0) | 61.4 (3.5) | 61.7 (3.0) | 60.9 (3.0) |
| OGBN ARXIV | GCN | **55.7 (0.2)** | 54.2 (0.5) | 32.1 (18.5) | 12.2 (9.0) | 7.2 (0.3) | 7.0 (0.8) |
| | GCN + PAIRNORM | 60.2 (0.3) | 60.3 (0.3) | **61.0 (0.2)** | 59.1 (0.3) | 53.2 (2.8) | 11.8 (4.7) |
| | GCNII | 58.2 (0.3) | **58.5 (0.6)** | 56.4 (2.5) | 56.9 (0.3) | 56.6 (0.4) | 56.2 (0.2) |
| | GATv2 + PAIRNORM | **60.2 (0.1)** | 59.4 (0.1) | 59.2 (0.1) | 56.7 (0.4) | OOM | OOM |
| | GRAPHSAGE + PAIRNORM | 60.6 (0.1) | 60.6 (0.2) | **60.7 (0.2)** | 58.1 (0.7) | 30.2 (11.9) | 15.8 (7.7) |
| PHOTO | GCN | **86.0 (4.5)** | 73.7 (8.8) | 48.6 (10.0) | 29.9 (12.1) | 22.8 (4.4) | 25.9 (2.0) |
| | GCN + PAIRNORM | **87.0 (1.1)** | 78.0 (2.2) | 81.7 (2.2) | 81.4 (2.0) | 78.7 (5.3) | 65.0 (6.2) |
| | GCNII | **86.9 (2.0)** | 86.5 (1.7) | 85.5 (3.1) | 84.5 (4.5) | 85.0 (3.8) | 85.1 (5.1) |
| | $G^2$ | 78.3 (6.2) | 77.4 (10.7) | 82.9 (3.3) | **83.6 (3.2)** | 82.3 (3.3) | 77.2 (4.0) |
| | GRAPHSAGE | **82.4 (3.9)** | 80.5 (3.4) | 54.9 (16.0) | 38.7 (13.7) | 24.0 (4.3) | 24.0 (4.3) |
| | GRAPHSAGE + PAIRNORM | **75.8 (3.1)** | 58.5 (5.4) | 69.3 (12.6) | 74.7 (3.4) | 51.9 (12.7) | 52.2 (19.1) |
| | $G^2$ (GRAPHSAGE) | 80.5 (6.2) | **83.7 (6.0)** | 82.5 (5.9) | 80.2 (9.4) | OOM | OOM |
| COMPUTERS | GCN | **77.7 (3.7)** | 73.1 (4.5) | 54.4 (16.3) | 21.1 (14.7) | 24.4 (16.8) | 10.7 (12.0) |
| | GCN + PAIRNORM | **80.7 (2.4)** | 75.3 (3.2) | 76.0 (2.3) | 65.3 (17.8) | 45.5 (20.0) | 59.2 (11.1) |
| | GCNII | **81.5 (1.9)** | 81.3 (1.5) | 81.2 (1.7) | 75.1 (14.5) | 80.2 (3.9) | 81.2 (2.1) |
| | $G^2$ | 66.1 (9.9) | **74.6 (4.9)** | 73.3 (7.9) | 70.6 (9.8) | 74.5 (3.7) | OOM |
| | GRAPHSAGE | **81.4 (1.7)** | 77.4 (4.1) | 30.4 (20.6) | 32.1 (15.3) | 9.9 (5.5) | 7.3 (4.8) |
| | GRAPHSAGE + PAIRNORM | 80.7 (2.3) | **81.4 (2.1)** | 74.8 (6.1) | 56.5 (21.5) | 38.2 (19.4) | 42.1 (14.6) |
| | $G^2$ (GRAPHSAGE) | 73.6 (6.3) | 70.3 (8.1) | **75.0 (3.8)** | 73.2 (9.4) | 67.3 (14.5) | OOM |

*Table 7.* Full table of experiments on graph-level tasks with all tested models, comprising GIN and GATv2 and GraphSAGE in combination with tested mitigation techniques for over-squashing for a range of model depths. The accuracy (* or average precision) is given in % with the standard deviation in brackets. The best performing model is highlighted in bold. The optimal depth per model and dataset is underlined.

| DATASET | MODEL | LAYERS | | | | | |
|---|---|---|---|---|---|---|---|
| | | **2** | **4** | **8** | **16** | **32** | **64** |
| MUTAG | GIN | 87.7 (3.1) | 86.3 (3.3) | 84.2 (2.2) | 86.7 (3.0) | 84.4 (3.9) | 88.2 (3.7) |
| | EGP (GIN) | **91.1 (1.4)** | 74.6 (11.2) | 61.1 (8.0) | 65.8 (2.3) | 63.9 (6.1) | 66.0 (3.5) |
| | GIN + BORF | 88.1 (2.7) | 86.5 (2.0) | 86.1 (1.7) | 87.0 (3.9) | 83.2 (4.9) | 83.5 (5.2) |
| | GIN + SDRF | 89.5 (2.6) | 86.3 (3.0) | 81.9 (3.0) | 82.3 (7.3) | 88.1 (3.9) | 86.5 (3.8) |
| | GATv2 | 88.1 (1.0) | 89.6 (3.5) | 90.4 (4.3) | 76.8 (14.9) | 63.7 (12.4) | 48.8 (13.9) |
| | EGP (GATv2) | 83.7 (3.4) | 86.8 (3.9) | 66.0 (8.6) | 57.0 (12.0) | 56.5 (13.4) | 63.5 (5.2) |
| | GATv2 + BORF | 70.7 (6.5) | 82.8 (3.3) | 80.4 (5.0) | 73.0 (11.2) | 68.8 (12.3) | 63.5 (5.2) |
| | GATv2 + SDRF | 75.3 (6.3) | 87.2 (2.7) | 80.4 (6.8) | 77.2 (8.0) | 65.3 (18.2) | 69.5 (9.3) |
| | GRAPHSAGE | 87.0 (1.4) | 88.9 (1.5) | 87.0 (3.8) | 80.5 (8.2) | 63.2 (7.9) | 68.4 (0.0) |
| | EGP (GRAPHSAGE) | 88.2 (1.0) | 88.6 (1.2) | 78.9 (4.8) | 68.4 (5.1) | 54.2 (10.4) | 48.6 (16.4) |
| | GRAPHSAGE + BORF | 85.4 (2.0) | 85.6 (2.4) | 81.6 (6.7) | 73.0 (8.7) | 65.8 (9.5) | 68.2 (0.7) |
| | GRAPHSAGE + SDRF | 88.1 (3.0) | 86.0 (2.9) | 85.3 (6.1) | 80.9 (6.7) | 72.8 (5.6) | 72.8 (2.2) |
| ENZYMES | GIN | **60.6 (3.4)** | 59.0 (1.4) | 56.8 (3.2) | 47.7 (4.3) | 31.0 (3.6) | 17.7 (2.5) |
| | EGP (GIN) | 50.2 (2.7) | 45.3 (3.8) | 24.7 (3.5) | 18.8 (4.4) | 17.9 (3.3) | 16.6 (0.4) |
| | GIN + BORF | 56.9 (2.4) | 58.4 (2.1) | 57.6 (2.0) | 47.1 (3.0) | 30.5 (3.7) | 19.1 (2.9) |
| | GIN + SDRF | 50.3 (2.6) | 48.4 (2.3) | 44.6 (2.7) | 35.9 (3.0) | 27.0 (3.1) | 16.4 (1.8) |
| | GATv2 | 36.3 (2.8) | 37.7 (2.8) | 30.9 (2.6) | 23.3 (3.0) | 16.7 (0.4) | 16.7 (0.0) |
| | EGP (GATv2) | 33.1 (3.1) | 38.5 (2.2) | 27.6 (3.6) | 16.6 (2.5) | 17.0 (0.6) | 16.7 (0.1) |
| | GATv2 + BORF | 37.4 (4.1) | 33.5 (2.1) | 31.3 (2.9) | 24.1 (2.8) | 17.1 (0.8) | 16.7 (0.0) |
| | GATv2 + SDRF | 33.7 (2.7) | 35.3 (3.1) | 30.4 (3.5) | 23.5 (3.3) | 18.1 (2.6) | 16.7 (0.0) |
| | EGP (GRAPHSAGE) | 29.0 (1.8) | 42.7 (3.3) | 34.2 (2.8) | 22.3 (2.8) | 17.4 (1.1) | 16.8 (0.2) |
| | GRAPHSAGE + BORF | 28.6 (1.6) | 31.0 (2.5) | 30.3 (2.8) | 25.6 (3.0) | 16.7 (0.0) | 16.7 (0.0) |
| | GRAPHSAGE + SDRF | 36.4 (1.8) | 38.7 (2.2) | 32.9 (3.4) | 23.3 (3.4) | 18.5 (1.7) | 16.7 (0.0) |
| PROTEINS | GIN | 64.0 (1.1) | 64.3 (1.3) | 64.2 (1.3) | 63.9 (1.2) | 66.5 (2.2) | 66.1 (1.9) |
| | EGP (GIN) | 61.8 (0.7) | 65.6 (2.2) | 65.7 (3.4) | 68.1 (8.6) | 41.0 (15.5) | 46.8 (18.2) |
| | GIN + BORF | 65.3 (1.2) | 63.7 (1.0) | 72.6 (1.9) | 65.6 (2.1) | 66.9 (2.1) | 68.0 (1.4) |
| | GIN + SDRF | 70.3 (0.9) | 72.0 (1.3) | **72.9 (1.8)** | 72.8 (2.0) | 71.4 (2.0) | 68.1 (1.9) |
| | GATv2 | 66.9 (0.7) | 67.7 (0.6) | 69.0 (0.7) | 66.8 (1.5) | 59.7 (5.5) | 57.2 (2.8) |
| | EGP (GATv2) | 65.4 (1.9) | 70.1 (1.9) | 66.9 (4.5) | 59.6 (10.7) | 58.5 (1.8) | 59.8 (0.0) |
| | GATv2 + BORF | 67.2 (0.5) | 67.3 (0.8) | 69.4 (1.5) | 62.1 (7.7) | 52.3 (7.4) | 48.0 (4.6) |
| | GATv2 + SDRF | 66.6 (0.5) | 68.3 (0.3) | 69.8 (0.7) | 66.7 (5.0) | 58.6 (7.8) | 57.2 (3.7) |
| | GRAPHSAGE | 68.1 (0.5) | 69.4 (0.6) | 68.5 (0.5) | 70.1 (0.5) | 61.8 (2.3) | 59.8 (0.0) |
| | EGP (GRAPHSAGE) | 67.6 (0.2) | 63.2 (1.4) | 55.2 (3.7) | 54.8 (4.7) | 55.8 (5.0) | 43.7 (5.7) |
| | GRAPHSAGE + BORF | 67.9 (0.3) | 68.5 (0.5) | 70.3 (0.9) | 68.5 (2.0) | 61.1 (2.7) | 59.8 (0.0) |
| | GRAPHSAGE + SDRF | 67.3 (0.4) | 69.7 (0.6) | 67.1 (0.4) | 66.7 (1.3) | 62.2 (3.9) | 59.9 (0.0) |
| PEPTIDES-FUNC | GIN | 59.3 (0.5) | **60.7 (1.0)** | 57.9 (1.1) | 60.7 (1.1) | 55.7 (1.2) | 42.2 (8.5) |
| | EGP (GIN) | 41.6 (0.3) | 42.9 (0.6) | 21.4 (3.2) | 17.5 (0.2) | 17.3 (0.3) | 17.2 (0.4) |
| | GIN + BORF | 59.0 (0.8) | 59.7 (0.6) | 57.9 (1.4) | 55.8 (1.4) | 49.7 (0.8) | 47.0 (1.4) |
| | GIN + SDRF | 56.7 (0.6) | 55.2 (0.8) | 53.1 (0.7) | 53.8 (1.4) | 49.8 (1.5) | 40.4 (4.1) |
| | GATv2 | 41.2 (2.4) | 46.6 (1.1) | 47.8 (2.7) | 32.0 (7.6) | 17.8 (1.2) | 17.4 (1.3) |
| | EGP (GATv2) | 38.0 (1.3) | 41.6 (1.3) | 22.2 (6.6) | 17.3 (0.3) | 16.9 (1.0) | 17.3 (0.5) |
| | GATv2 + BORF | 43.4 (0.6) | 45.5 (1.8) | 45.5 (1.8) | 30.5 (7.7) | 17.6 (0.4) | 17.3 (1.0) |
| | GATv2 + SDRF | 39.9 (0.7) | 41.4 (1.1) | 43.3 (3.0) | 32.0 (5.8) | 17.7 (1.0) | 17.6 (1.0) |
| | GRAPHSAGE | 45.2 (0.2) | 48.5 (0.9) | 51.9 (2.4) | 26.0 (10.2) | 18.5 (1.4) | 17.7 (1.2) |
| | EGP (GRAPHSAGE) | 42.4 (0.3) | 42.8 (0.3) | 41.3 (0.4) | 20.7 (7.6) | 17.1 (0.5) | 17.0 (0.7) |
| | GRAPHSAGE + BORF | 47.4 (0.5) | 50.1 (0.8) | 48.1 (1.5) | 38.4 (7.0) | 24.6 (0.0) | 24.6 (0.0) |
| | GRAPHSAGE + SDRF | 44.2 (0.2) | 45.8 (1.0) | 46.3 (1.5) | 40.7 (4.0) | 29.0 (2.0) | 26.0 (3.6) |

*Table 8.* Full table of experiments on node-level tasks with all tested models, comprising GCN and GATv2 and GraphSAGE in combination with tested mitigation techniques for over-squashing for a range of model depths. The accuracy is given in % with the standard deviation in brackets. The best performing model is highlighted in bold. The optimal depth per model and dataset is underlined.

| DATASET | MODEL | LAYERS | | | | | |
|---|---|---|---|---|---|---|---|
| | | 2 | 4 | 8 | 16 | 32 | 64 |
| CORA | GCN | 77.1 (3.3) | 74.3 (0.4) | 36.0 (12.4) | 31.9 (0.0) | 31.9 (0.0) | 31.9 (0.0) |
| | EGP (GCN) | 77.7 (4.7) | 31.9 (0.0) | 31.9 (0.0) | 31.9 (0.0) | 31.9 (0.0) | 31.9 (0.0) |
| | GCN + BORF | 75.4 (0.3) | 71.3 (6.9) | 35.9 (11.9) | 31.9 (0.0) | 31.9 (0.0) | 31.9 (0.0) |
| | GCN + SDRF | 76.2 (3.0) | 36.1 (12.6) | 34.2 (7.0) | 31.9 (0.0) | 31.9 (0.0) | 31.9 (0.0) |
| | GATv2 | 75.5 (2.8) | 73.9 (0.8) | 31.9 (0.0) | 31.9 (0.0) | 31.9 (0.0) | 31.9 (0.0) |
| | EGP (GATv2) | 78.3 (4.4) | 31.9 (0.0) | 31.9 (0.0) | 31.9 (0.0) | 31.9 (0.0) | 31.9 (0.0) |
| | GATv2 + BORF | 76.0 (3.2) | 74.1 (0.6) | 36.0 (12.3) | 31.9 (0.0) | 31.9 (0.0) | 31.9 (0.0) |
| | GATv2 + SDRF | 74.4 (0.5) | 61.7 (15.9) | 35.4 (7.5) | 31.9 (0.0) | 31.9 (0.0) | 31.9 (0.0) |
| | GRAPHSAGE | **82.1 (5.1)** | 74.3 (0.5) | 31.9 (0.0) | 31.9 (0.0) | 31.9 (0.0) | 31.9 (0.0) |
| | EGP (GRAPHSAGE) | 78.0 (4.6) | 31.9 (0.0) | 31.9 (0.0) | 31.9 (0.0) | 31.9 (0.0) | 31.9 (0.0) |
| | GRAPHSAGE + BORF | 76.1 (0.9) | 71.8 (13.7) | 35.1 (9.7) | 31.9 (0.0) | 31.9 (0.0) | 31.9 (0.0) |
| | GRAPHSAGE + SDRF | 79.6 (5.4) | 71.1 (13.6) | 52.0 (15.5) | 31.9 (0.0) | 31.9 (0.0) | 31.9 (0.0) |
| CITESEER | GCN | 75.5 (0.8) | 74.5 (0.7) | 73.5 (0.4) | 58.1 (12.3) | 53.6 (19.2) | 18.1 (0.0) |
| | EGP (GCN) | 75.7 (0.6) | 63.8 (4.8) | 42.6 (2.0) | 20.5 (4.4) | 18.5 (1.5) | 18.7 (2.0) |
| | GCN + BORF | 75.9 (0.5) | 74.5 (0.7) | 73.4 (0.7) | 69.7 (4.7) | 57.5 (14.0) | 24.3 (9.2) |
| | GCN + SDRF | 76.1 (0.6) | 74.8 (0.7) | 73.4 (0.6) | 66.4 (5.5) | 52.0 (18.9) | OOM |
| | GATv2 | 75.0 (0.6) | 74.1 (0.8) | 72.9 (0.6) | 58.7 (11.5) | 39.9 (8.4) | 21.1 (2.4) |
| | EGP (GATv2) | 75.6 (0.5) | 73.9 (0.8) | 60.7 (10.2) | 26.1 (6.3) | 18.6 (1.5) | 18.1 (0.0) |
| | GATv2 + BORF | 74.9 (0.5) | 74.4 (0.8) | 73.2 (0.7) | 56.4 (12.9) | 44.3 (14.7) | 18.6 (1.5) |
| | GATv2 + SDRF | 74.7 (0.7) | 74.1 (0.8) | 73.0 (0.8) | 58.0 (13.0) | 37.3 (10.7) | 18.6 (1.5) |
| | GRAPHSAGE | 75.5 (0.4) | 74.1 (0.4) | 72.8 (0.6) | 66.2 (4.7) | 49.9 (16.3) | 18.8 (0.0) |
| | EGP (GRAPHSAGE) | **76.4 (0.5)** | 73.3 (0.9) | 61.0 (7.7) | 47.3 (11.3) | 24.6 (5.3) | 24.2 (3.0) |
| | GRAPHSAGE + BORF | 75.7 (0.5) | 73.9 (0.7) | 72.4 (0.9) | 62.8 (9.9) | 40.2 (18.7) | 18.8 (0.0) |
| | GRAPHSAGE + SDRF | 75.7 (0.5) | 73.8 (0.6) | 72.4 (0.4) | 67.1 (9.0) | 40.2 (21.3) | 19.4 (0.0) |
| PUBMED | GCN | 87.7 (0.4) | 86.1 (0.2) | 83.0 (4.2) | 79.3 (5.6) | 63.2 (11.8) | 40.5 (0.4) |
| | EGP (GCN) | 87.1 (0.4) | 86.8 (0.4) | 69.1 (9.5) | 40.5 (0.4) | 40.4 (0.2) | 40.4 (0.3) |
| | GCN + BORF | 87.7 (0.4) | 85.9 (0.4) | 84.0 (3.2) | 75.9 (12.7) | 68.6 (7.8) | 40.4 (0.6) |
| | GCN + SDRF | 87.7 (0.3) | 86.0 (0.4) | 85.0 (0.4) | 77.9 (5.1) | 47.0 (12.5) | 40.6 (0.4) |
| | GATv2 | OOM | 86.7 (0.8) | 84.7 (0.3) | 73.6 (2.0) | 63.1 (14.7) | 40.7 (0.0) |
| | EGP (GATv2) | 87.9 (0.2) | 87.9 (0.5) | 73.8 (1.3) | 40.7 (0.0) | 40.7 (0.0) | 40.7 (0.0) |
| | GATv2 + BORF | 87.8 (0.4) | 86.5 (0.5) | 83.6 (3.0) | 78.4 (4.6) | 54.8 (14.2) | 41.0 (0.3) |
| | GATv2 + SDRF | 87.6 (0.4) | 86.9 (0.4) | 83.6 (3.3) | 71.3 (10.5) | 52.7 (14.4) | 40.9 (0.3) |
| | GRAPHSAGE | 88.3 (0.4) | **88.6 (0.4)** | 81.4 (5.8) | 75.2 (3.8) | 40.7 (0.0) | 40.8 (0.2) |
| | EGP (GRAPHSAGE) | 88.2 (0.3) | 86.7 (4.1) | 70.2 (10.0) | 53.3 (15.0) | 54.4 (12.3) | 54.7 (12.0) |
| | GRAPHSAGE + BORF | 87.1 (0.3) | 88.5 (0.5) | 79.2 (6.9) | 73.5 (16.9) | 40.7 (0.0) | 40.8 (0.2) |
| | GRAPHSAGE + SDRF | 88.2 (0.5) | 87.9 (0.4) | 82.1 (7.0) | 72.8 (0.6) | 40.7 (0.0) | 40.7 (0.0) |
| ROMAN-EMPIRE | GCN | 41.5 (0.7) | 27.3 (1.2) | 18.7 (0.6) | 16.5 (0.3) | 15.0 (0.9) | 14.6 (0.6) |
| | EGP (GCN) | 41.9 (0.2) | 21.9 (0.6) | 14.1 (0.2) | 14.3 (0.4) | 14.6 (0.9) | 14.7 (1.1) |
| | GCN + BORF | 40.9 (0.7) | 26.7 (0.7) | 18.8 (0.7) | 16.5 (0.3) | 15.1 (0.5) | 14.4 (0.6) |
| | GCN + SDRF | 41.6 (0.7) | 27.3 (1.1) | 18.5 (0.5) | 16.4 (0.3) | 15.1 (0.7) | 14.7 (1.2) |
| | GATv2 | 39.2 (2.1) | 38.5 (2.8) | 34.4 (3.7) | 19.3 (3.5) | 14.2 (0.3) | 13.9 (0.1) |
| | GATv2 + BORF | 40.5 (1.5) | 38.2 (2.6) | 35.9 (3.3) | 20.9 (2.9) | 14.1 (0.4) | 13.9 (0.1) |
| | GATv2 + SDRF | 39.6 (1.6) | 37.7 (1.9) | 35.3 (3.9) | 20.3 (3.6) | 14.1 (0.3) | 13.9 (0.0) |
| | GRAPHSAGE | 64.6 (0.3) | 61.5 (2.6) | 46.5 (3.6) | 33.2 (3.1) | 13.9 (0.1) | 13.9 (0.1) |
| | EGP (GRAPHSAGE) | **66.4 (0.4)** | 58.3 (3.6) | 45.9 (4.0) | 20.8 (6.4) | 13.9 (0.0) | 14.0 (0.1) |
| | GRAPHSAGE + BORF | 63.4 (0.8) | 61.5 (2.9) | 49.4 (3.8) | 29.5 (6.4) | 13.9 (0.1) | 13.9 (0.0) |
| | GRAPHSAGE + SDRF | 64.6 (0.2) | 62.7 (1.7) | 48.5 (3.1) | 30.3 (6.0) | 13.8 (0.2) | 13.9 (0.1) |
| OGBN ARXIV | GCN | 55.7 (0.2) | 54.2 (0.5) | 32.1 (18.5) | 12.2 (9.0) | 7.2 (0.3) | 7.0 (0.8) |
| | EGP (GCN) | 57.6 (0.3) | 23.8 (10.3) | 6.7 (0.4) | 7.8 (0.3) | 7.2 (0.7) | 6.9 (1.2) |
| | EGP (GATv2) | 57.1 (0.4) | 55.6 (1.0) | 21.7 (18.0) | 5.9 (0.0) | 5.9 (0.0) | OOM |
| | EGP (GRAPHSAGE) | **57.7 (0.5)** | 51.7 (6.1) | 44.2 (8.7) | 16.5 (11.0) | 20.3 (1.0) | 20.1 (1.4) |
| PHOTO | GCN | 86.0 (4.5) | 73.7 (8.8) | 48.6 (10.0) | 29.9 (12.1) | 22.8 (4.4) | 25.9 (2.0) |
| | EGP (GCN) | 77.9 (13.1) | 56.2 (15.9) | 32.1 (9.2) | 24.3 (1.8) | 25.1 (1.7) | 27.1 (3.0) |
| | GCN + BORF | 82.0 (5.7) | 71.9 (14.9) | 53.1 (15.4) | 31.1 (8.1) | 28.9 (12.0) | 25.9 (1.3) |
| | GCN + SDRF | 85.8 (4.1) | 79.1 (6.4) | 53.3 (16.3) | 35.1 (13.4) | 29.0 (8.8) | 26.0 (1.3) |
| | GRAPHSAGE | 82.4 (3.9) | 80.5 (3.4) | 54.9 (16.0) | 38.7 (13.7) | 24.0 (4.3) | 24.0 (4.3) |
| | EGP (GRAPHSAGE) | 81.2 (1.0) | 76.5 (5.0) | 54.0 (12.7) | 26.6 (5.5) | 25.8 (0.4) | 25.3 (0.9) |
| | GRAPHSAGE + BORF | **86.6 (1.7)** | 73.7 (6.4) | 55.9 (22.9) | 35.6 (17.0) | 24.6 (1.9) | 23.0 (5.7) |
| | GRAPHSAGE + SDRF | 85.2 (1.0) | 74.7 (16.6) | 60.5 (19.1) | 30.8 (12.6) | 26.1 (0.1) | 24.3 (4.3) |
| COMPUTERS | GCN | 77.7 (3.7) | 73.1 (4.5) | 54.4 (16.3) | 21.1 (14.7) | 24.4 (16.8) | 10.7 (12.0) |
| | EGP (GCN) | 81.3 (1.6) | 73.5 (1.8) | 11.6 (9.8) | 3.3 (2.4) | 3.9 (2.2) | 3.9 (3.2) |
| | GCN + SDRF | 76.1 (2.6) | 72.4 (4.2) | 64.5 (4.6) | 25.4 (17.9) | 26.1 (18.6) | 9.6 (11.0) |
| | GRAPHSAGE | **81.4 (1.7)** | 77.4 (4.1) | 30.4 (20.6) | 32.1 (15.3) | 9.9 (5.5) | 7.3 (4.8) |
| | EGP (GRAPHSAGE) | 78.0 (2.4) | 74.8 (3.6) | 30.2 (14.5) | 9.9 (9.1) | 6.9 (6.5) | 6.7 (3.1) |
| | GRAPHSAGE + SDRF | 81.2 (1.5) | 75.7 (4.6) | 51.2 (17.5) | 16.6 (16.1) | 8.4 (5.5) | 11.8 (10.3) |

