# OpenReview forum: "Position: Don't be Afraid of Over-Smoothing and Over-Squashing"
_ICML.cc/2026/Position_Paper_Track — Submitted to ICML 2026 Position Paper Track_

### Official Review · Reviewer_pr7W · 2026-03-11

**Significance:** 3
**Argument Clarity:** 3
**Rating:** 5
**Confidence:** 4

**Questions:**

Q1: Could the authors consider evaluating additional smoothness-related metrics to further support the claim that over-smoothing plays a negligible role in practice?

Q2: Given the paper’s position that over-smoothing and over-squashing may not be the main concerns in many settings, could the authors discuss the potential limitations of this view and identify scenarios where these phenomena might still be important?

Q3: The “Call to Action” section still does not deliver a sufficiently clear message to the community, especially to researchers working on over-smoothing and/or over-squashing in GNNs, regarding what they should focus on next, why these directions matter, and how such efforts should be carried out.

====================after rebuttal==============================

The authors have addressed the questions and key concerns raised in my comments.

**Alternative Views Section:**

Yes

**Compliance With Llm Reviewing Policy A Conservative:**

Affirmed.

**Discussion Potential:**

3

**Final Justification:**

After checking the authors' rebuttal, I am satisfied with the additional clarification and explanation. I have changed my score from 4 to 5.

**Paper Summary:**

This position paper argues that current GNN research may place too much emphasis on over-smoothing and over-squashing as the main explanations for performance degradation. Instead, the authors suggest that performance often deteriorates because the model’s receptive field simply does not contain sufficiently useful information. Through a series of experiments, they show that this lack of informative local signals can better explain the observed decline, and further note that, for many real-world datasets, large receptive fields and long-range dependency modeling are not always necessary. Based on these observations, the paper advocates that future theoretical work on GNNs should pay closer attention to the locality and decomposability of label information, and develop corresponding statistical measures to better guide model design and research directions.

**Position:**

Yes

**Position In Title:**

Yes

**Related Work:**

3

**Strengths And Weaknesses:**

Strengths:

1. The paper is clearly structured and easy to follow, and the arguments are presented in a coherent and well-organized manner.

2. The concern about the practical impact of over-smoothing and over-squashing in GNNs is interesting and worth discussing.

3. The experimental study is strong, and the results are generally sufficient to support the main observations of the paper.

Weaknesses:

1. The relationship between over-smoothing and over-squashing is still not entirely clear; for example, it would be helpful to discuss whether they are coupled phenomena, whether they influence each other, or whether they should be viewed as largely separate issues.

2. If the paper argues that over-smoothing is not the main problem in practice, then it would be useful to better explain the effectiveness of commonly used techniques such as skip connections, which are often understood as alleviating over-smoothing and do appear to help empirically.

3. The paper questions the role of over-squashing, but the issue of how over-squashing should be properly measured or diagnosed is still not fully clarified.

**Support:**

3

---

> ### Author Rebuttal · Authors · 2026-03-31
>
> We are grateful for your exhaustive and in-depth review. We address your raised weaknesses and questions point-by-point in what follows.
>
> **Weakness 1:**
> Indeed, this is an interesting topic. There exists good theoretical research about the relationship between the two phenomena of over-smoothing and over-squashing. For example, Nguyen et al. [1] relate both phenomena to positive and negative graph curvature, and Giraldo et al. [2] investigate the relationship between the two based on the spectral gap of the graph Laplacian. Both state that there is a trade-off between over-smoothing and over-squashing when it comes to graph rewiring to bridge structural bottlenecks. We are happy to include these two papers in our manuscript and would be grateful if the reviewer could point us towards other related work in this direction. However, we do not see how this discussion affects our position and wouldn’t necessarily call this incompletely understood relationship a weakness of our manuscript. We focus on the consequences of these phenomena in real-world datasets rather than on their, possibly common, cause.
>
> **Weakness 2:**
> We conduct a comprehensive analysis of a variety of commonly used mitigation techniques in our manuscript. In particular, in Section 3.1, we explain what approaches exist in the literature and how these techniques work. Skip connections specifically are part of the GCNII architecture, which we also included in our experiments. The observed empirical effectiveness of skip-connections can be conjectured to arise as a consequence of a variety of benefits they are known to convey. Chief among these is the fact that enable the model to more strongly focus on the representations obtained from early, small receptive fields, which aligns with the hypothesis we formulate in our paper that most learning tasks on graphs are local in nature. However, in general, it is well-known that skip-connections can enable improved gradient flow and are known to boost performance in a variety of deep learning contexts, including computer vision. We believe that precisely attributing the performance improvements of these methods is out of scope of our position paper. Our work studies the effectiveness of these methods in alleviating the phenomena of over-smoothing and over-squashing.
>
> **Weakness 3:**
> We fully agree with your assessment that it is unclear, up until now, how over-squashing can be measured, which serves as strong motivation for our call to action. We state that it would be beneficial to more carefully study the real-world impact of the theoretical phenomena we attempt to resolve in our literature.
>
> **Question 1:**
> The evaluation of all prominently established methods to measure over-smoothing is part of this work. As we explain in Section 3.1, some smoothness measures are not applicable to the GCN. In Appendix A, we provide further details on the measures we use in this paper, as well as further results for experiments with GATv2 and GraphSAGE message passing methods. Overall, we included two versions of the Dirichlet energy, MAD and the node similarity metric by Wu [3] in our experiments. We would be grateful if the reviewer could point us to any specific over-smoothing metrics that they feel are missing from our work.
>
> **Question 2:**
> Our position is mainly limited by the range of datasets that are currently established in the graph learning literature. As we point out in our introduction, we expect new datasets and new learning tasks to emerge over the next years in the pursuit to increase the relevance of our research field. It is possible, but based on current empirical evidence unlikely, that over-smoothing or over-squashing become more relevant in these datasets.
>
> **Question 3:**
> Our core message to researchers in the field of graph learning is to empirically measure the impact of theoretical phenomena (like over-smoothing and over-squashing) in real-world datasets. Making presumed problems measurable in practice is crucial to ensure the relevance of theoretical research for transformative real-world applications. We argue that especially the literature over-smoothing and over-squashing would benefit from more extensive empirical investigation into these phenomena, with the aim of finding problems that drive model performance in practice.
>
>
>
> [1] Nguyen, K., Nong, H., Nguyen, V., Ho, N., Osher, S., and Nguyen, T. Revisiting over-smoothing and over-squashing using Ollivier-Ricci curvature. In the International Conference on Machine Learning (ICML), 2023.
>
> [2] J. Giraldo, K. Skianis, T. Bouwmans and F. D. Malliaros. On the Trade-off between Over-smoothing and Over-squashing in Deep Graph Neural Networks. ACM International Conference on Information and Knowledge Management, 2023.
>
> [3] Wu, X., Ajorlou, A., Wu, Z., and Jadbabaie, A. Demystifying oversmoothing in attention-based Graph Neural Networks. In Advances in Neural Information Processing Systems (NeurIPS), 2023.

---

> > ### Author Rebuttal · Reviewer_pr7W · 2026-04-02
> >
> > The authors have addressed my concerns and I am happy to change my score accordingly.

---

### Official Review · Reviewer_JrPw · 2026-03-11

**Significance:** 2
**Argument Clarity:** 2
**Rating:** 4
**Confidence:** 3

**Questions:**

see weakness

**Alternative Views Section:**

Yes

**Compliance With Llm Reviewing Policy A Conservative:**

Affirmed.

**Discussion Potential:**

3

**Paper Summary:**

This position paper challenges the prevailing focus in Graph Neural Network (GNN) research on the phenomena of over-smoothing and over-squashing, arguing that these issues are less critical in practical applications than commonly assumed. The authors posit that performance drops in deep GNNs typically stem from uninformative receptive fields rather than over-smoothing, a claim they support with extensive empirical evidence showing that architectural techniques designed to mitigate over-smoothing fail to yield significant performance gains or justify deeper models. Furthermore, they contest the universally detrimental view of over-squashing by suggesting that relevant information in real-world graphs is frequently localized within small neighborhoods, which questions the necessity of capturing long-range interactions across entire receptive fields. Ultimately, the authors advocate for a paradigm shift in the field, urging researchers to move away from these theoretical bottlenecks and instead utilize statistics to measure the true localization and factorization of label-relevant information in graph datasets.

**Position:**

Yes

**Position In Title:**

Yes

**Related Work:**

3

**Strengths And Weaknesses:**

Strengths:
1. Extensive Empirical Validation: The authors back their position with a comprehensive suite of experiments testing exponentially increasing model depths from 2 to 64 layers. They evaluate multiple architectures designed to mitigate over-smoothing (such as GCN with PairNorm, GCNII, and $G^2$) and over-squashing (such as EGP, BORF, and SDRF).
2. Decoupling Metrics from Performance: A major strength of the experimental design is demonstrating the disconnect between theoretical metrics and predictive accuracy. The paper shows that while architectural interventions successfully stabilize theoretical measures like Mean Average Distance (MAD) and Dirichlet energy , optimal model depths remain shallow and overall accuracy does not significantly improve.

Weakness:
1. Reliance on Traditional Benchmarks: To support the claim that over-smoothing and over-squashing are practically irrelevant, the authors test on datasets like Cora, CiteSeer, and PubMed. These traditional node-classification benchmarks are notoriously small and homophilous, often requiring only a 1- or 2-hop radius to solve. While the authors do include more complex datasets like Roman-Empire and Peptides-func, leaning heavily on datasets known to lack long-range dependencies naturally biases the results away from needing deep GNNs.
2. Lack of Formal Metrics for the Alternative Hypothesis: The paper repeatedly suggests that "uninformative receptive fields" are to blame for performance drops. However, the authors do not propose, define, or calculate a specific metric to quantify this phenomenon within their experiments. They leave the creation of these statistics as a "Call to Action" for future research , making their core counter-hypothesis somewhat abstract.

**Support:**

2

---

> ### Author Rebuttal · Authors · 2026-03-31
>
> Thank you very much for your thoughtful comments. In what follows, we address the two raised weaknesses.
>
> **Weakness 1:**
> We agree with your statement that most node-classification benchmarks require a small receptive field. We conducted further experiments on an additional dataset of the Long Range Graph Benchmark, which we present in our response to Reviewer fCaG. These additional results and the results in our manuscript show that the statement about the small problem radius is not only true for the datasets Cora, CiteSeer and PubMed, but also for a wide range of established benchmark datasets. Especially, as many impactful publications on over-smoothing [1, 2, 3] and over-squashing [2, 4, 5] work with traditional datasets like Cora, CiteSeer and PubMed as well as Enzymes and Proteins, we view this extensive suite of experiments as a suitable foundation for commenting on this literature. In general, we want to highlight the scarcity of real-world long-range graph benchmark datasets, making it somewhat difficult to extensively lean on such data.
>
> **Weakness 2:**
> The development of specific metrics, as called for in our call for action, is where we believe one should distinguish a position paper from a technical paper. Both of the approaches we suggest in the paper, to measure the localisation and factorisation of the feature and label distribution would require technical papers to be fully fleshed out. Our submission aims to contribute a new perspective on the two extensively investigated phenomena, over-smoothing and over-squashing, by putting the theoretical work into the context of realistic empirical experimentation. However, our call to action, while heavily leaning on these two specific examples, can also be relevant to other “real-world learning problems” as we call them in the conclusion of our paper. The fundamental idea of our position is stated in our introduction “A phenomenon becomes a problem when its impact can be measured”. We believe our position to be a valuable contribution to the current graph learning community in itself and to be best published in a position paper.
>
>
> [1] Kipf, T. N. and Welling, M. Semi-supervised classification with graph convolutional networks. In International Conference on Learning Representations (ICRL), 2017.
>
> [2] Chen, D., Lin, Y., Li, W., Li, P., Zhou, J., and Sun, X. Measuring and relieving the over-smoothing problem for Graph Neural Networks from the topological view. In
> AAAI Conference on Artificial Intelligence, pp. 3438–3445, 2019.
>
> [3] Keriven, N. Not too little, not too much: a theoretical analysis of graph (over)smoothing. In Advances in Neural Information Processing Systems (NeurIPS), 2022.
>
> [4] Alon, U. and Yahav, E. On the bottleneck of graph neural networks and its practical implications. In International Conference on Learning Representations (ICRL), 2020.
>
> [5] Topping, J., Giovanni, F. D., Chamberlain, B. P., Dong, X., and Bronstein, M. M. Understanding over-squashing and bottlenecks on graphs via curvature. In International Conference on Learning Representations (ICRL), 2022.

---

> > ### Author Rebuttal · Reviewer_JrPw · 2026-04-04
> >
> > For the second weakness, the authors’ rebuttal makes little sense. It is exactly important for a position paper  to exclusively make the hypothesis clear, or otherwise the whole position wouldn’t stand. If the basis of the position remains ambiguous, the position itself, either correct or wrong, will be of little research impact.

---

### Official Review · Reviewer_fCaG · 2026-03-12

**Significance:** 2
**Argument Clarity:** 3
**Rating:** 2
**Confidence:** 5

**Questions:**

The author claim that
> , problems that justify the consideration of a very large receptive field per node, one may almost surely want to explore alternatives such as Graph Transformers

Why does the author make such claim? There is no evidence that the graph transformers perform better than GNNs on these problems. There are also several issues of the current grpah transformers.

**Alternative Views Section:**

Yes

**Compliance With Llm Reviewing Policy A Conservative:**

Affirmed.

**Discussion Potential:**

2

**Paper Summary:**

The authors claim that addressing over-smoothing and over-squashing maybe not necessary for the real-world graphs. The authors benchamrk over-smoothing and over-squashing methods on several benchmark datasets and conclude the claims.

**Position:**

Yes

**Position In Title:**

Yes

**Related Work:**

3

**Strengths And Weaknesses:**

Strengths:

1. The authors conduct several experiments on widely used benchmark datasets to support their claim.

2. The papers is well-written and easy to follow.


Weaknesses:

1. It is well known that many commonly used benchmark datasets mainly rely on local information for prediction. As a result, phenomena such as over-smoothing and over-squashing may not become the key factors causing performance degradation on these datasets. The authors may add more experiments on the long-range benchamrks.

2. In this paper, however, the analysis appears to be heavily limited to a small set of such commonly used benchmarks. Therefore, the conclusions about the practical importance of over-smoothing and over-squashing may be overly dependent on the specific properties of these datasets. It remains unclear whether the same observations would hold in more diverse or real-world graph settings. This is also one of the reasons why many prior studies on over-smoothing and over-squashing rely on synthetic datasets.

In summary, I did not find the main claims or observations in this paper to be particularly surprising or novel.

**Support:**

2

---

> ### Author Rebuttal · Authors · 2026-03-31
>
> Thank you very much for your attentive review. We respond to your observed weaknesses and question in the following.
>
> **Weaknesses:**
> Your assessment that in most benchmark datasets, relevant information is present locally, fully aligns with the position we take in our paper, that in most currently established learning tasks, the problem radius is relatively small. Our paper already contains results for the dataset Pepdtides-func of the Long Range Graph Benchmark (LRGB), which supports our claims. Further, we report additional supportive results on the Peptides-struct dataset, which is also part of LRGB, in this rebuttal. LRGB is currently the only collection of real-world datasets established in the literature with proclaimed long-range dependencies. Indeed, as you say, other benchmarks for long-range interaction rely on synthetic data or synthetic labels, which allows for the isolated study of certain well-specified phenomena, as we also laude in Section 4.3 of our paper. However, synthetic data cannot form the basis of statements on the real-world impact of the studied phenomena and is therefore out of scope for our position paper. We advocate for measuring the practical relevance of theoretically studied phenomena to guide further theoretical research on problems that occur in real-world applications.
>
> We want to respectfully disagree with the statement that our conclusions are based on a “small set of such commonly used benchmarks”. In our paper, we provide experiments on 11 real-world datasets, which is on-par with or exceeding many technical papers in our field. However, in response to your comment, we have conducted experiments on a twelfth dataset (see the table below) from the LRGB benchmark, demonstrating that our conclusions hold for datasets that are presumed to contain long-range interactions.
>
> | Dataset | Model | 2 | 4 | 8 | 16 | 32 | 64 |
> | --- | --- | --- | --- | --- | --- | --- | --- |
> | Peptides-struct | GIN | 0.294 (0.003) | 0.290 (0.005) | **_0.272 (0.002)_** | 0.274 (0.003) | 0.287 (0.006) | 0.378 (0.002) |
> | Peptides-struct | EGP (GIN) | _0.438 (0.004)_ | 0.446 (0.006) | 0.537 (0.075) | 0.651 (0.010) | 0.655 (0.005) | 0.667 (0.002) |
> | Peptides-struct | GIN + BORF | _0.274 (0.001)_ | 0.276 (0.004) | 0.275 (0.010) | 0.289 (0.008) | 0.293 (0.007) | 0.331 (0.010) |
> | Peptides-struct | GIN + SDRF | 0.315 (0.002) | 0.305 (0.003) | _0.304 (0.003)_ | 0.312 (0.010) | 0.325 (0.007) | 0.382 (0.010) |
>
> We expect that for a significant proportion of the graph learning community, our findings are surprising, as many works on over-smoothing [1, 2, 3] and over-squashing [2, 4, 5] are based on the benchmark datasets we consider in our position paper. Especially, the result that when models are trained and carefully hyperparameter tuned, we observe stable Dirichlet energy measurements in Figures 1-4 of the paper, is surprising in the context of the line of reasoning of other works [6,7], where the Dirichlet energy was argued to decay exponentially in all datasets to motivate inevitable over-smoothing.
>
> **Question:**
> In the claim you quote, we state that one would almost surely want to explore working with Graph Transformers if a large receptive field per node is to be considered. This claim is trivially true in our view: In each layer of a standard GNN, the receptive field of a node is limited to its neighbours in the graph, whereas in a standard Graph Transformer layer, the receptive field of each node contains the entire graph. Note that our claim is limited in scope to these facts, and we do not aim to convey that Graph Transformers are optimal model choices in general.
>
> [1] Kipf, T. N. and Welling, M. Semi-supervised classification with graph convolutional networks. In International Conference on Learning Representations (ICRL), 2017.
>
> [2] Chen, D., Lin, Y., Li, W., Li, P., Zhou, J., and Sun, X. Measuring and relieving the over-smoothing problem for graph neural networks from the topological view. In
> AAAI Conference on Artificial Intelligence, pp. 3438–3445, 2019.
>
> [3] Keriven, N. Not too little, not too much: a theoretical analysis of graph (over)smoothing. In Advances in Neural Information Processing Systems (NeurIPS), 2022.
>
> [4] Alon, U. and Yahav, E. On the bottleneck of graph neural networks and its practical implications. In International Conference on Learning Representations (ICRL), 2020.
>
> [5] Topping, J., Giovanni, F. D., Chamberlain, B. P., Dong, X., and Bronstein, M. M. Understanding over-squashing and bottlenecks on graphs via curvature. In International Conference on Learning Representations (ICRL), 2022.
>
> [6] Rusch, T. K., Bronstein, M. M., and Mishra, S. A survey on oversmoothing in graph neural networks. In arXiv, 2023a.
>
> [7] Park, M., Choi, S., Heo, J., Park, E., and Kim, D. The oversmoothing fallacy: A misguided narrative in GNN research. In arXiv, 2025.

---

> > ### Author Rebuttal · Reviewer_fCaG · 2026-04-01
> >
> > Thanks for the authors' rebuttal. However, the oversmoothing and oversquashing can happen in the realworld applications, for example [1] cliams that the oversquashing might be the issue for the difficulty of learning GNN-based SAT solvers. They should depend on specific tasks and data. Therefore, the conclusion in the paper is too strong and only based on a few datasets.
> >
> >
> > [1] A Geometric Perspective on the Difficulties of Learning GNN-based SAT Solvers

---

### Official Review · Reviewer_aR2G · 2026-03-13

**Significance:** 3
**Argument Clarity:** 4
**Rating:** 5
**Confidence:** 4

**Questions:**

The paper correctly highlights a trend in the GRL community. Even if the paper presents a point of view that is at least partially agreeable, I believe it is worth noting that a purely theoretical analysis of GRL remains valuable, especially in the academic community. This might represent an alternative perspective worth mentioning. Nevertheless, I acknowledge that the paper does not oppose such theoretical research, but rather calls for a way to quantify its impact.

**Alternative Views Section:**

Yes

**Compliance With Llm Reviewing Policy A Conservative:**

Affirmed.

**Discussion Potential:**

3

**Final Justification:**

The paper offers a critique of the strong focus, within a large portion of the graph learning community, on theoretical issues such as over-squashing and over-smoothing. It provides some evidence and invites the community to better assess their actual impact on real-world applications. As a position paper, I find that it could serve as a valuable starting point for the community to reflect on the problems that truly matter in graph learning.

**Paper Summary:**

The paper criticizes the current focus of graph representation learning research on theoretical aspects such as over-squashing and over-smoothing. In particular, the paper argues that, even if interesting from a theoretical perspective, these problems have limited practical impact on real-world datasets. These claims are supported by an experimental evaluation of most known methods that directly tackle these problems, showing that:

- most datasets present localized relevant information; as such, shallow GNNs are sufficient to achieve good performance;
- in the few cases where a larger receptive field is beneficial (i.e., more convolutional layers), the gain in performance of deeper convolutions is negligible with respect to the computational overhead;
- over-squashing is, in general, not problematic for real-world datasets, suggesting a possible factorization of relevant features.

The paper calls for the community to build statistical tools to measure the relevance of such theoretical limitations in real-world datasets as a strategy to guide future theoretical developments toward more relevant directions.

**Position:**

Yes

**Position In Title:**

Yes

**Related Work:**

4

**Strengths And Weaknesses:**

This position paper puts a lens on the trends in graph representation learning, focusing on theoretical aspects that often have limited practical impact. The paper is well structured and properly introduces the problem and experimentally supports its claims.

The call to action that asks for statistical tools to measure the relevance of these theoretical aspects in real-world datasets is an interesting perspective on how to evaluate their practical impact.

Some claims, such as the exchange of information between different communities and, more generally, the limited relevance of over-squashing, would benefit from further discussion.

**Support:**

3

---

> ### Author Rebuttal · Authors · 2026-03-31
>
> Thank you very much for your careful and particularly insightful review. In what follows, we address the stated weakness and the raised question.
>
> **Weakness:**
> We agree that our claims on over-squashing and especially the topic of the inter-community exchange of information merit further discussion in our manuscript. Upon consideration, we would like to add the following, more direct formulation of our understanding.
> The presence of bottleneck edges indicates limited information exchange between structural communities. The phenomenon of bottleneck edges only becomes a problem if this inter-community information exchange is required for the learning task. We argue that such inter-community exchange is not productive in most practical applications. Our rationale is that working with graphs implies some relevance of the relational structure of the nodes. As this structure contains information about communities, this information should be label-relevant, and most nodes in a community will have the same label. Bridging these bottlenecks would be unproductive in these cases. The results from Table 8 of our paper support this argument, as in most cases, the rewiring techniques do not yield better performance. We will add this more direct formulation of our view to our manuscript.
>
> **Question:**
> Thank you very much for this comment. We completely agree. In this position paper, we do not want to criticise theoretical work in our literature in general and explicitly want to encourage targeted theoretical work. We believe that purely theoretical work is among the greatest strengths of our community. In our manuscript, we attempted to convey this understanding, for example:
> - at the beginning of Section 3.2, where we highlight the benefits of theoretical work: “The extensive work on over-smoothing has led to a good theoretical understanding of this phenomenon”.
> - We call for further theoretical work in Section 4 on Page 5: “We believe that continued theoretical work on over-squashing should be guided by measurable problems that limit the performance of GNNs in practical applications”, and on Page 7 in our call to action: “...to guide further theoretical research efforts…”.
>
> Our position is that the allocation of research efforts should be guided towards problems that can be measured in real-world applications. We will additionally clarify in our manuscript that our position is not aiming to curtail theoretical research purely to problems of immediate measurable real-world impact, but rather that such measures would be a useful guide for future theoretical research in our community. Therefore, we do not see the relevance of theoretical research as an alternative perspective, but as very much aligned with our position. We will include this view more explicitly in our revised manuscript.

---

> > ### Author Rebuttal · Reviewer_aR2G · 2026-04-03
> >
> > I thank the authors for their clarifications. I still believe that the over-squashing intuition might not be generalizable to all networks or tasks.  For instance, while a note-level prediction (as in table 8) might mostly be influenced by intra-community information, other tasks, such as fraud detection, might need visibility into both intra- and inter-community information exchange.

---

### Decision · Program_Chairs · 2026-04-30

**Decision:**

Reject

**Comment:**

The reviews were somewhat mixed. The reviewers noted several strong points of the paper but also identified weaknesses that need to be addressed.

Major Strong points:

* The paper is well structured and properly introduces the problem and experimentally supports its claims. The paper is easy to follow.
* The authors conduct several experiments on widely used benchmark datasets to support their claim.
* The concern about the practical impact of over-smoothing and over-squashing in GNNs is interesting and worth discussing.

Major Weak Points:

* The Support and Significance scores are low in multiple reviews.
* None of the reviewers fully championed the paper (No strong accept recommendations)
* The paper suggests that "uninformative receptive fields" are to blame for performance drops. However, the authors do not propose, define, or calculate a specific metric to quantify this phenomenon within their experiments.  Without proper definitions, the position is somewhat ambiguous.

Therefore, the paper cannot be accepted to the conference at this time.
We encourage the authors to revise and resubmit to an appropriate future venue.

The AC has read the authors’ rebuttals and comments and has incorporated them into the decision-making process.